



# Seasonal mass variations show timing and magnitude of meltwater storage in the Greenland ice sheet

Jiangjun Ran[1], Miren Vizcaino[1], Pavel Ditmar[1], Michiel R. van den Broeke[2], Twila Moon[3], Christian R. Steger[2], Ellyn M. Enderlin[4], Bert Wouters[2], Brice Noël[2], Catharina H. Reijmer[2], Roland Klees[1], and Min Zhong[5]

[1]Department of Geoscience and Remote Sensing, Delft University of Technology, Delft, The Netherlands
[2]Institute for Marine and Atmospheric Research, Utrecht University, Utrecht, The Netherlands
[3]National Snow and Ice Data Center, Cooperative Institute for Research in Environmental Sciences, University of Colorado, Boulder, CO, USA
[4]Climate Change Institute and School of Earth and Climate Science, University of Maine, Orono, USA
[5]State Key Laboratory of Geodesy and Earth's Dynamics, Institute of Geodesy and Geophysics, Chinese Academy of Sciences, Wuhan, China

*Correspondence to:* J. Ran (j.ran@tudelft.nl)

**Abstract.** The Greenland Ice Sheet (GrIS) is currently losing ice mass as the result of changes in the complex ice-climate interactions that have been driven by global climate change. In order to accurately predict future sea level rise, the mechanisms driving the observed mass loss must be better understood. Here, we combine data from the satellite gravimetry mission GRACE, surface mass balance (SMB) output of RACMO 2.3, and ice discharge estimates to analyze the mass budget of Greenland at

5   various temporal and spatial scales. Firstly, in agreement with previous estimates, we find that the rate of mass loss from Greenland observed by GRACE was between -277 and -269 Gt/yr in 2003-2012. This estimate is consistent with the sum of individual contributions: surface mass balance (SMB, around $216 \pm 122$ Gt/yr) and ice discharge ($520 \pm 31$ Gt/yr), indicating a good performance of the regional climate model. Secondly, we examine the average accelerations of mass anomalies in Greenland over 2003-2012, suggesting that the SMB ($-23.3 \pm 2.7$ Gt/yr$^2$) contributes 75% to the total acceleration observed by

10   GRACE. The remaining contributions to the mass loss acceleration for entire Greenland are statistically insignificant. Finally and most importantly, this study suggests the existence of a substantial meltwater storage during summer, with a peak value of 80-120 Gt in July. The robustness of this estimate is demonstrated by using both different GRACE-based solutions and different meltwater runoff estimates (namely, RACMO 2.3 and SNOWPACK). Meltwater storage in the ice sheet occurs primarily due to storage in the high-accumulation regions of the southeast (SE) and northwest (NW) parts of Greenland. Analysis of seasonal

15   variations in outlet glacier discharge shows that the contribution of ice discharge to the observed signal is minor (at the level of only a few Gt) and does not explain the intra-annual differences between the total mass and SMB signals.

## 1 Introduction

During the last decade (2006–2015), the Greenland Ice Sheet (GrIS) has been rapidly losing mass, contributing on average 0.9 (0.4–1.2) mm/yr to global mean sea level rise (Enderlin et al., 2014; Stocker et al., 2013; van den Broeke et al., 2016). The





Gravity Recovery And Climate Experiment (GRACE) mission is a powerful tool to monitor ice mass variations in Greenland (including both GrIS and peripheral glaciers), from the monthly to multi-year time scales. After removing the impacts of Glacial Isostatic Adjustment (GIA) and ocean, those mass variations (referred as MB in Eq. 1) measured by GRACE result from the combination of three effects: (i) mass variations reflecting the surface mass balance (SMB), (ii) mass removal due

to ice discharge (ID) to the ocean, and (iii) mass variations ($\Delta m$) which include all processes not related to SMB and ice discharge, for instance, en- and sub-glacial meltwater storage (MS):

$$MB = SMB - ID + \Delta m. \tag{1}$$

Recent GrIS mass loss has been quantified in several studies (e.g., Shepherd et al., 2012; Schrama et al., 2014; Velicogna et al., 2014). Furthermore, several authors have estimated the contribution to this mass loss from SMB and ice discharge

individually (van den Broeke et al., 2009; Enderlin et al., 2014; Velicogna et al., 2014; van den Broeke et al., 2016). To quantify the contribution of SMB, regional climate models (RCMs) are typically used, such as the Regional Atmospheric Climate Model v. 2 (RACMO2) (Ettema et al., 2009), MAR (Fettweis et al., 2005) and Hirham (Christensen et al., 1996). The annual ice discharge rates are estimated by combining ice flow velocity data and ice thickness data at the flux gates (Thomas et al., 2000). Importantly, ice flow velocities have demonstrated an overall increase during the last decade that is superimposed

by inter-annual variability (Moon et al., 2012), so that they have to be monitored on a regular basis.

The analysis of GrIS mass variations at the intra-annual time scale is still limited. This is largely because (i) the accuracy and spatial resolution of GRACE monthly solutions is relatively poor, as compared to long-term trend estimates, and (ii) ice velocity data at this time scale are scarce (typically, only a few estimates per year are available, often spanning only a few years). A first attempt to combine GRACE data and SMB modelling in order to evaluate an ice dynamics model of the GrIS at

the monthly time scale was made by Schlegel et al. (2016). Alexander et al. (2016) examined spatial patterns of GrIS seasonal mass variations using a regional climate model, an ice sheet model and GRACE. The only study of multi-regional intra-annual variations of GrIS outlet glacier velocities was conducted by Moon et al. (2014), who analyzed 55 marine-terminating glaciers in northwest and southeast Greenland over the period 2009–2013.

The GrIS mass balance is also characterized by supra-, en- and subglacial meltwater retention. An example is the abundance

of supra-glacial lakes primarily in west Greenland, which store water during the melt season (Selmes et al., 2011). Sub-glacial hydrology is an area of active research (see e.g., Chandler et al., 2013; Slater et al., 2015). Until now, however, time-varying en- and sub-glacial meltwater retention (included in $\Delta m$ of Eq. 1) is poorly quantified and mostly investigated at a local scale. For instance, Rennermalm et al. (2013) quantified meltwater retention in a small watershed (36-65 km$^2$) near Kangerlussuaq. They suggested that ∼54% of liquid water is retained for one to six months in this watershed. To date, only one study has

utilized GRACE data to quantify meltwater retention at ice-sheet-wide scales (Van Angelen et al., 2014). By fitting de-trended GRACE observations and SMB model output, it was found that the mean period of meltwater retention at the whole-ice-sheet scale is ∼18 days.





In this study, we analyze the individual mass variation contributors (see Eq. 1) to total inter- and intra-annual mass variations over Greenland at both regional and whole-ice-sheet scales. In particular, this study makes a first ice-sheet-wide attempt to quantify the amplitude and timing of meltwater storage. For this purpose, we combine observations of total mass variations from GRACE with observations of ice discharge to the ocean (Enderlin et al., 2014; Moon et al., 2014), as well as modelled

SMB output from RACMO2.3 (Noël et al., 2015) and SNOWPACK (Steger et al., 2017). Since the spatial resolution of GRACE data is limited, the obtained estimates cover both the GrIS and the parts of Greenland outside the GrIS, including the tundra and the peripheral glaciers disconnected from the GrIS.

In Sect. 2, we discuss the exploited methods and data. The results are presented and analyzed in Sect. 3. Finally, the conclusions are presented in Sect. 4.

## 2  Data and methods

### 2.1  GRACE

We use the 5th release of GRACE monthly gravity field solutions from the Center for Space Research (CSR) at the University of Texas as input to compute total mass variations. Each solution is provided as a set of spherical harmonic coefficients up to degree 96, and supplied with a full error covariance matrix. For the sake of consistency with previous GRACE-based estimates,

we limit the considered time interval to Jan. 2003 – Dec. 2013. Since data for 9 months are missing, 123 months in total are used. Furthermore, a reduced time interval (Jan. 2003 – Dec. 2012) is also considered in order to make the obtained estimates consistent with ice discharge data from Enderlin et al. (2014); see Sect 2.3. for a further discussion. Due to strong noise in the $C_{2,0}$ coefficients, we replaced them with available estimates based on satellite laser ranging (Cheng et al., 2013). The degree-one coefficients, which are missing in the GRACE products, are taken from Swenson et al. (2008). The GRACE solutions are

corrected for Glacial Isostatic Adjustment (GIA), which is triggered by a relief of ice load since the last glacial maximum, with the model from A et al. (2013).

To estimate mass variations over Greenland at a regional scale, we make use of a novel data processing methodology (Ran et al., 2017), which is based on the mascon approach. Greenland is split into 28 patches or "mascons" (Fig. 1), which are complemented by 9 patches outside Greenland to absorb signals from the surrounding areas. Temporal variations (anomalies)

of surface density (i.e., mass variation per unit area) within each patch are assumed to be spatially homogeneous. Thus, the total mass anomaly within each patch is just a product of surface density anomaly and patch area. Those mass anomalies are computed for each month independently, without any regularization. Ultimately, mass anomalies are summed up over individual Drainage Systems (DSs) or entire Greenland.

A further description of the adopted GRACE data processing methodology is provided in the appendix. In order to inves-

tigate the robustness of GRACE-based mass anomalies, we estimate them using different processing parameters. This leads to multiple sets of GRACE solutions: two primary ones (estimated with and without applying data weighting) and alternative ones. We consider also the estimates produced by other research teams: the JPL mascon solution by Watkins et al. (2015), the





CSR mascon solution by Save et al. (2016), the GSFC mascon solution by Luthcke et al. (2013), and the solution by Wouters et al. (2008).

Similar to van den Broeke et al. (2009), we aggregate the 28 mascons inside Greenland into five DSs. We refer to these DSs as: (a) North (N); (b) Northwest (NW); (c) Southeast (SE); (d) Southwest (SW); and (e) Northeast (NE) (Fig. 1). We slightly

5 shifted southwards the border between NW and SW DSs, as compared to van den Broeke et al. (2009), in order to ensure that the SW DS is mostly limited to land-terminating glaciers.

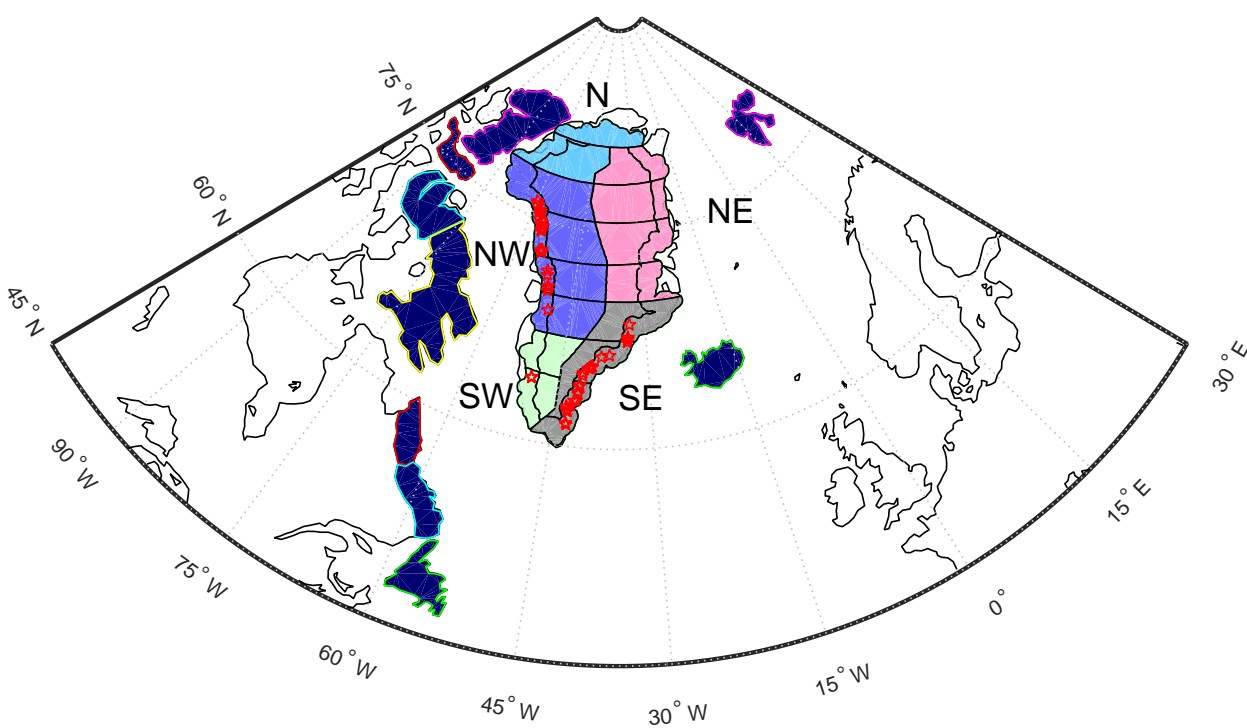

**Figure 1.** The 28-mascon parameterization of Greenland used in this study for GRACE data processing. For the purpose of further analysis, these patches are merged into five drainage systems (N, NE, SE, SW, and NW). 55 glaciers utilized to compute seasonal ice discharge variations are marked as red pentagrams.

## 2.2 SMB modeling

SMB values over 2003-2013 from two models, i.e., RACMO2.3 and SNOWPACK are analyzed. In both cases, the SMB estimates are obtained as the sum of individual SMB components:

10 $$\text{SMB} = \text{P} - \text{ES} + \text{SD} - \text{RU}. \qquad (2)$$

where P is precipitation, ES is evaporation/sublimation, SD is erosion/deposition due to snowdrift and RU is runoff. Note that the SMB output from RACMO2.3 is our primary choice due to its good performance (van den Broeke et al., 2009). Thus,



the SMB mass anomalies through this study are from RACMO2.3, unless stated otherwise. The output from SNOWPACK is included to evaluate the robustness of our findings with respect to the choice of SMB outputs simulated by different snow/firn models.

### 2.2.1 RACMO2.3

RACMO2.3 was developed by the Royal Netherlands Meteorological Institute (KNMI) and Institute for Marine and Atmospheric Research (IMAU), which is a part of Utrecht University in the Netherlands (Noël et al., 2015). RACMO2.3 provides daily SMB values with a spatial resolution of 11 km.

### 2.2.2 SNOWPACK

In addition to the daily SMB provided by RACMO2.3, we use SMB output from SNOWPACK (Steger et al., 2017). SNOW-
PACK is a state-of-the-art snow model that offers a more physically based snow densification scheme, the simulation of microstructural snow properties, and a higher near-surface vertical resolution, compared with the snow/firn module of RACMO2.3. A comparison of SNOWPACK with IMAU-FDM, a snow/firn model nearly identical to the one implemented in RACMO2.3, revealed a better performance of SNOWPACK for the GrIS, particularly for locations with comparably high amounts of liquid water input due to snow melt and rainfall (Steger et al., 2017). SNOWPACK was coupled offline to RACMO2.3 and run for
the full period 1960 – 2014 with the same surface mask (ocean, tundra, ice sheet) and horizontal resolution as RACMO2.3. As a consequence of forcing SNOWPACK with mass fluxes from RACMO2.3 at the snow-atmosphere interface, the first three components of Eq. 2 are identical for the two models. Differences in the simulated SMB are thus only caused by unequal partitioning of meltwater and rainfall into refreezing and runoff.

### 2.2.3 Processing the SMB outputs from RACMO2.3 and SNOWPACK

In this study, we integrate SMB outputs from RACMO2.3 or SNOWPACK in time, which results in cumulative values that can be interpreted as daily SMB mass anomalies. These values are averaged over monthly intervals for the sake of temporal consistency with the GRACE solutions. In order to make the SMB outputs (11-km resolution) spatially matching the GRACE resolution (around 300 km), we process them consistently with the GRACE data. More specifically, we converted the SMB outputs to gravity disturbances at the satellite altitude and limited their spectra to spherical harmonic degree/order 96. Then the
SMB per mascon is estimated by a least-square adjustment (see the Appendix).

Previous works on the sources of current GrIS mass loss used *relative* SMB outputs, i.e., anomalies with respect to an equilibrium state (1961–1990) (van den Broeke et al., 2009; Velicogna et al., 2014). Effectively, this means that the time-series of mass anomalies were de-trended to ensure that they are close to zero during the reference equilibrium period. In contrast, we use time-series of absolute SMB mass anomalies, i.e., without referring to a hypothesized equilibrium state. In this way, we
are able to extract more information from the datasets. For instance, absolute mass anomalies related to ice discharge cannot increase over time, which is a valuable constraint that facilitates the correct interpretation of the obtained results.





In addition, in contrast to the previous studies, we use SMB outputs that included non-GrIS areas of Greenland, too. This is because GRACE senses mass anomalies not only within the GrIS, but also at ice caps disconnected from the GrIS and tundra areas.

## 2.3 Ice discharge

We consider two different data sets. The first set was already presented in Enderlin et al. (2014). Here, it is used to reconstruct the 2003–2012 multi-year mass trends and accelerations, as well as to separate the contributions from SMB and ice discharge. It covers 178 outlet glaciers with annual resolution. Ice discharge observations of these glaciers are estimated by multiplying ice flow velocities with ice thickness values. Annual velocities are retrieved by means of feature tracking from (winter) Landsat 7 Enhanced Thematic Mapper Plus and the Advanced Spaceborne Thermal and Reflectance Radiometer (ASTER) data. Ice

discharge is calculated within 5 km of the grounding lines. The ice thicknesses at the flux gates are computed by subtracting bed elevations from surface elevations. Bed elevations are derived from NASA's Operation IceBridge airborne ice-penetrating radar data, whereas surface elevations are obtained from digital elevation models (Enderlin et al., 2014; Xu et al., 2015).

In addition, we produce the second data set, which is used to examine intra-annual variations of ice discharge. It covers 55 marine-terminating glaciers with sub-annual resolution for 2009–2013. The exploited ice flow velocities were obtained from

TerraSAR-X images delivered by the German Space Agency (DLR) (Moon et al., 2014). Ice thicknesses at the flux gates are interpolated from the IceBridge BedMachine Greenland version 2 data (Morlighem et al., 2015). Ice discharge (D) for a given glacier is defined as the ice mass flux across the flux gate (f) close to the glacier terminus (within ∼5 km):

$$D = \rho \int_f h(\boldsymbol{v} \cdot \boldsymbol{n})\mathrm{df}, \tag{3}$$

where $h$ is the ice thickness; $\boldsymbol{n}$ is the unit vector directed outwards normally to the flux gate; $\boldsymbol{v}$ is the ice flow velocity;

and $\rho$ is the ice density. When selecting flux gates, we paid attention to variations of the terminus position by checking the images of glaciers during the whole time interval, to make sure that the flux gate is in the upstream of the terminus all the time. Furthermore, a flux gate should span the whole outlet glacier to the ice flow edges. To compute $D$, we discretize the flux gates into nearly 200-m long intervals. The length of the last interval is adjusted to make sure that the ice flow edge is sampled. We then use the values ($h, \upsilon$ and $\boldsymbol{n}$) defined for the center of each interval, assuming that they are constant over the interval. Then

Eq. 3 becomes

$$D = \rho \sum_{i=1}^{N} d^i h^i (\boldsymbol{v^i} \cdot \boldsymbol{n}), \tag{4}$$

where N is the total number of intervals of the flux gate and $d^i$ is the length of the i-th interval.





## 3 Results and Discussion

### 3.1 Multi-year mass trend and acceleration budgets

First, we examine multi-year mass trends and accelerations in terms of the total mass balance and the contributions thereto from SMB and ice discharge. We approximate each mass anomaly time-series (cf. Fig. 2) with the following analytic function:

$$f(t) = a_1 + a_2(t - t_0) + a_3 \frac{(t - t_0)^2}{2} + a_4 \sin \omega t + a_5 \cos \omega t + a_6 \sin 2\omega t + a_7 \cos 2\omega t, \tag{5}$$

where $a_1, ..., a_7$ are parameters to be estimated, $t_0$ is a reference epoch defined as the middle of the considered time interval (i.e., July 1, 2008 for the time interval 2003-2013; January 1, 2008 for the time interval 2003-2012), and $\omega = 2\pi/T$ with $T = 1$ year.

In addition, we calculate the uncertainty of the trend estimate, $a_2$. This uncertainty of the GRACE-based estimate is composed of the error of the GIA model (we set it as 50% of the signal), the measurement errors of GRACE propagated from full variance-covariance matrix of monthly solutions, the uncertainty associated with a particular choice of the oceanic mascon layout (cf. Fig. A1), and signal leakage. The latter (including both signals which leaked from outside Greenland and signals from inside Greenland leaked between the mascons), was simulated numerically by defining the trend from ICESat as the reference. Note that the individual errors are summed up quadratically, by assuming that they are not correlated with each other. Unlike Velicogna and Wahr (2013), we do not consider errors from atmospheric and ocean circulation corrections, due to their small contribution. The similar approach is applied to estimate the uncertainty of acceleration (parameter $a_3$).

Our estimate of the total-mass linear trend, which is based on the primary GRACE data time-series produced with optimal data weighting, is -286 ± 21 Gt/yr for 2003-2013. The estimate obtained without data weighting is -279 ± 21 Gt/yr. The individual contributors to the errors in the trend estimates are shown in Table 1. Our estimates are in agreement with those published earlier for the time interval 2003-2013: -280 ± 58 Gt/yr (Velicogna et al., 2014) and -278 ± 19 Gt/yr (Schrama et al., 2014).

**Table 1.** Contribution of different error sources to the error in the total GrIS mass trend estimated from GRACE data both with and without data weighting, in Gt/yr).

| Contributor | Signal leakage | GIA correction | Ocean parameterization | GRACE data | Total error |
|---|---|---|---|---|---|
| Error | 15 | 8 | 7 | 10 | 21 |

We examine also the SMB and ice discharge contributions to the total mass trend. In this case, we consider the reduced time interval, 2003-2012, in order to be consistent with the ice discharge record, which ends in 2012 (Table 2). The multi-year average mass gains from SMB (RACMO2.3) over that period processed consistently with GRACE via data weighting and without data weighting are 216 ± 122 Gt/yr and 214 ± 122 Gt/yr, respectively. The uncertainty is computed by assuming a 9%





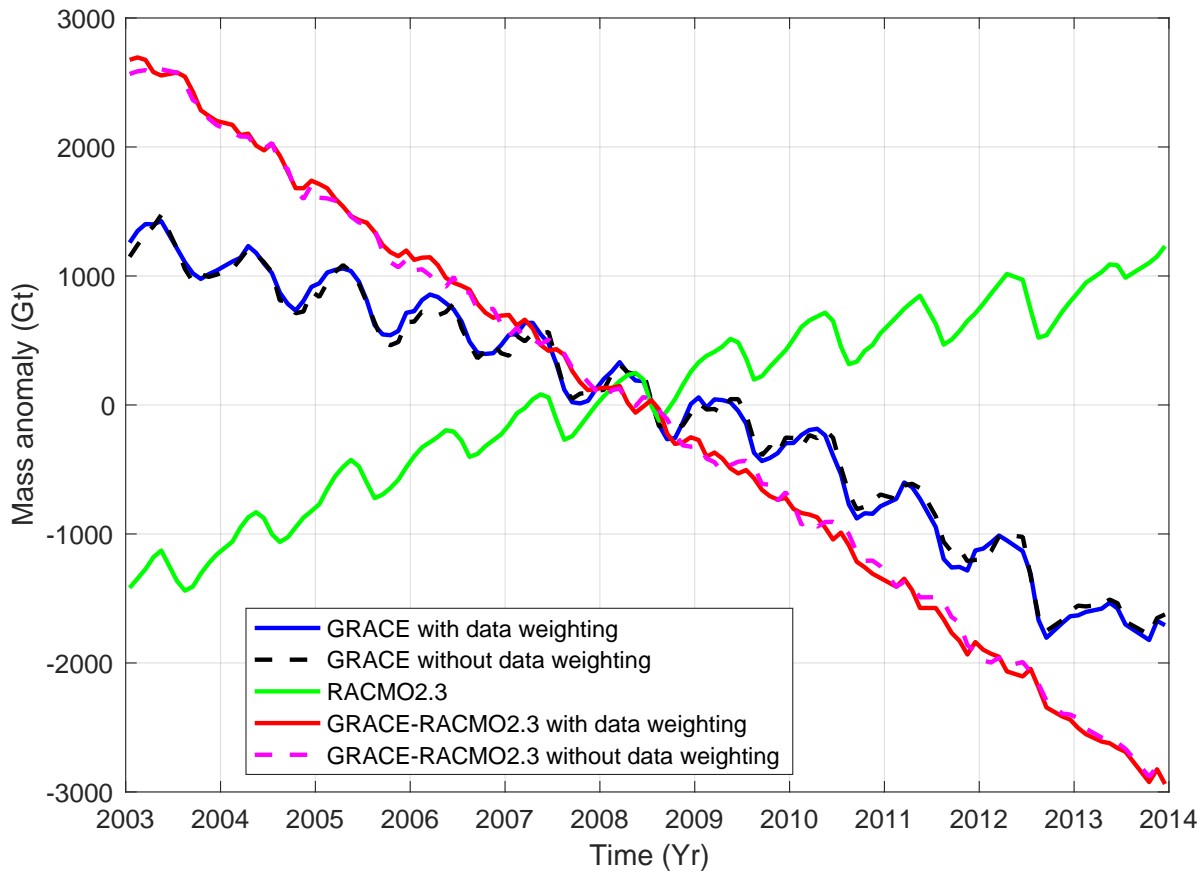

**Figure 2.** Time-series of mass anomalies over the period 2003-2013 for the entire GrIS: total mass anomalies from GRACE produced with/without data weighting (solid blue and dashed black, respectively), cumulative SMB anomalies from RACMO2.3 (green), and the difference between them, the "Total-SMB" residuals (solid red and dashed pink, respectively).

error in accumulation and a 15% error in meltwater runoff signals modeled by RACMO2.3, which is the typical uncertainty of RACMO2.3 (van den Broeke et al., 2016). The time-series of cumulative mass anomalies due to ice discharge and other processes not related to SMB is obtained as the difference between the total mass variations and the cumulative SMB-related ones; this difference will be referred to as the "Total-SMB" residuals ("Total minus SMB", cf. red and pink curves in Fig.

5  2). The associated rates of linear changes over 2003–2012 estimated with and without data weighting are 493 ± 124 Gt/yr and 483 ± 124 Gt/yr, respectively. Those estimates agree with the ice discharge estimate from Enderlin et al. (2014), 520 ± 31 Gt/yr. Notice that the GRACE time-series obtained with the optimal weighting seems to show a smoother behavior than that produced without data weighting. This is consistent with the analysis presented in Ran et al. (2017), who found that data





weighting substantially reduces the level of random noise in the GRACE GrIS mass change time-series.

Next, we present the results of a similar analysis for the individual DSs. The greatest total mass losses are observed by GRACE in DSs NW and SE (cf. Fig. 3 and Table 2). These two DSs account for ∼73-76% of the total mass loss over Green-

land, depending on whether data weighting is applied or not. The inter-annual behavior of these DSs is, however, different. SE loses mass with an approximately constant rate over the whole considered period. In contrast, NW is relatively stable over the period 2003-2005, but starts losing mass thereafter. The remaining three DSs lose mass at much smaller rates. Remarkably, two of these DSs (N and SW) show a similar behavior: they are relatively stable over the period 2003-2009, but start losing mass in 2010. These findings are consistent with Velicogna et al. (2014). The SMB is negative in two DSs (N and SW)

(cf. Table 2). However, with a large fraction of land-terminating glaciers, ice losses from ice discharge are an order of magnitude lower there than in the NW and SE DSs (cf. Fig. 3), resulting in only modest total mass loss in spite of the negative SMB.

The long-term trends of Total-SMB residuals in the DSs of NW, NE, and SW are consistent with the ice discharge estimates from Enderlin et al. (2014) within the error bar (Table 2). This suggests robustness of RACMO2.3 long-term SMB trends

there, under the assumption that the meltwater storage signal is mainly seasonal. In the SE and N, however, we find relatively large discrepancies between the Total-SMB residuals and ice discharge observations from Enderlin et al. (2014). Under the conditions of realistic GRACE error estimates and minimal multi-year meltwater storage, all these inconsistencies suggest a precipitation overestimation in the SE and underestimation in the N in RACMO2.3. However, it is also possible to explain large discrepancies for the SE drainage system by inaccurate ice discharge estimates there due to various sources of large uncertain-

ties: ice velocities (due to a decorrelation of SAR images in the presence of fast ice flows), ice thicknesses, and corrections for SMB signals at the locations between flux gates and grounding line.

Average accelerations of mass anomalies over the period 2003-2012 are also estimated using Eq. 5 (parameter $a_3$). Over the entirety of Greenland, the SMB (-23.3±2.7 Gt/yr$^2$) contributes with 75% to the total acceleration observed by GRACE

(Table 3). This is close to the estimates of Velicogna et al. (2014), who assessed the contribution of SMB to the total GrIS mass loss acceleration as 79%. The contribution of the Total-SMB residuals to the mass loss acceleration for entire Greenland is statistically insignificant. Analysis of individual drainage systems leads to similar conclusions (cf. Table 3).

### 3.2 Seasonal mass variations

We analyze the mean annual cycles of total (GRACE) and cumulative SMB (RACMO2.3) mass anomalies over the period

2003-2013 (Fig. 4). To derive them, we divide the entire period into eleven overlapping 13-month time intervals, each of which starts in December of the previous year and ends in December of the current year. Then, the mean mass anomaly for each calendar month is estimated by linear regression, together with one bias parameter per time interval, which accounts for a long-term variability. This scheme is less sensitive to gaps in data time-series than the plain averaging of mass anomalies per calendar month. The uncertainties of mean mass anomalies of GRACE are propagated from errors in each monthly GRACE estimate.





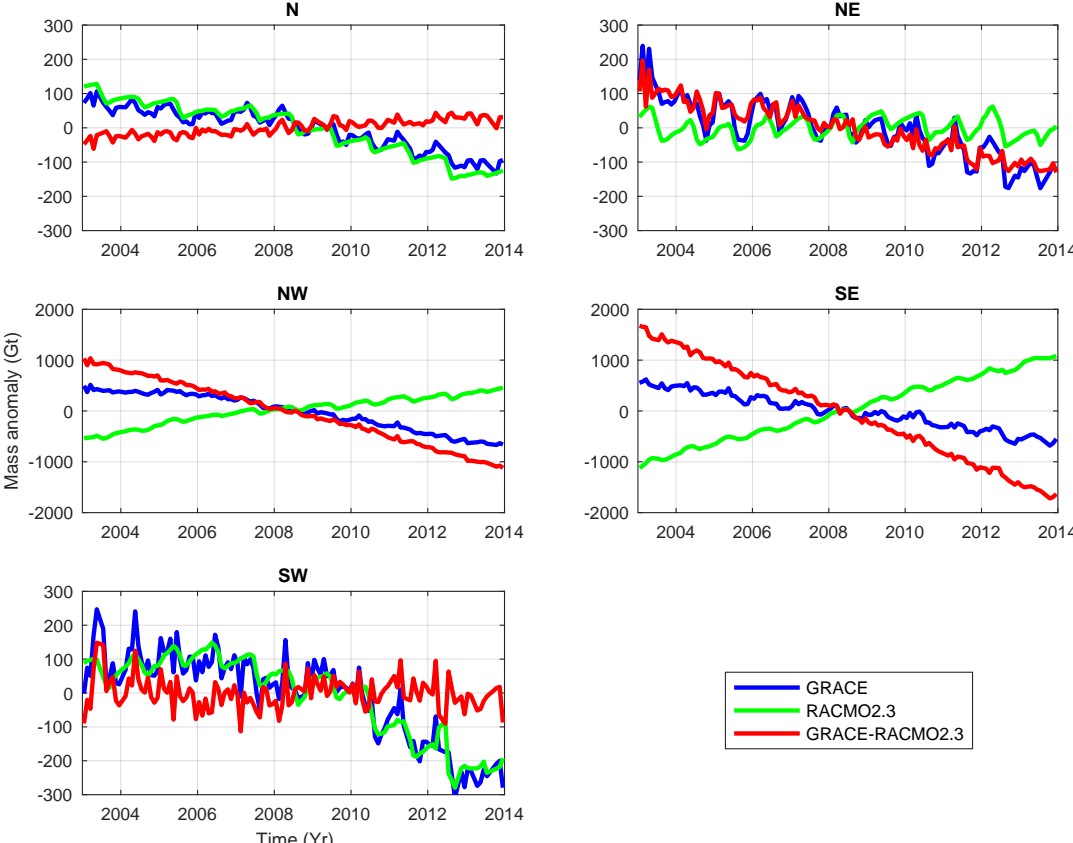

**Figure 3.** Same as Fig. 2, but for the GRACE-based estimates produced with data weighting at the drainage system scale. The estimates produced without data weighting are not shown as they are very similar to those with data weighting.

The uncertainties of cumulative SMB mean mass anomalies are computed by assuming 9% and 15% errors in modeled mean mass anomalies due to precipitation and runoff, respectively, as suggested by van den Broeke et al. (2016). The uncertainties of the Total-SMB residuals are the root-sum-square of the standard deviations of noise in GRACE and cumulative SMB estimates.

5    The whole-Greenland mean annual cycles of total and cumulative SMB mass anomalies present smooth month-to-month variations (Fig. 4). Importantly, the estimates of both types refer to the mean values for the months considered. The total mass anomaly from GRACE reaches its maximum in March and then steadily decreases until September. The most rapid mass loss is observed in July-August (∼200 Gt). In contrast, the cumulative SMB decreases over a much shorter period - only from May





**Table 2.** Linear mass change rates over the period 2003-2012 for individual drainage systems and entire Greenland: Total, SMB-related, and the Total-SMB residuals (GRACE minus SMB), as well as ice discharge (Gt/yr). The sign of Total-SMB residuals is changed to make them directly comparable with ice discharge estimates.

| Contributor | Data weighting | N | NW | NE | SW | SE | GrIS |
|---|---|---|---|---|---|---|---|
| Area ($10^{12}$m$^2$) | - | 0.26 | 0.69 | 0.60 | 0.21 | 0.40 | 2.16 |
| Total (GRACE) | Yes | -16 ± 11 | -106 ± 23 | -20 ± 16 | -29 ± 10 | -105 ± 23 | -277 ± 21 |
| Total (GRACE) | No | -18 ± 11 | -99 ± 23 | -26 ± 16 | -31 ± 10 | -96 ± 23 | -269 ± 21 |
| SMB | Yes | -24 ± 11 | 78 ± 28 | 1 ± 20 | -36 ± 27 | 197 ± 39 | 216 ± 122 |
| SMB | No | -24 ± 11 | 76 ± 28 | 3 ± 20 | -43 ± 27 | 202 ± 39 | 214 ± 122 |
| -(Total-SMB) | Yes | -8 ± 16 | 184 ± 36 | 21 ± 26 | -7 ± 29 | 302 ± 45 | 493 ± 124 |
| -(Total-SMB) | No | -6 ± 16 | 175 ± 36 | 29 ± 26 | -12 ± 29 | 298 ± 45 | 483 ± 124 |
| Ice discharge | - | 21 ± 13 | 206 ± 14 | 41 ± 10 | 18 ± 7 | 234 ± 20 | 520 ± 31 |

**Table 3.** Acceleration of mass change over the period 2003-2012 for individual drainage systems and entire Greenland: total, SMB-related, and the Total-SMB residuals (GRACE minus SMB), as well as ice discharge (Gt/yr$^2$). The sign of Total-SMB residuals is changed to make them directly comparable with ice discharge estimates.

| Contributor | Data weighting | N | NW | NE | SW | SE | GrIS |
|---|---|---|---|---|---|---|---|
| Total (GRACE) | Yes | -2.9 ± 1.5 | -15.6 ± 3.1 | -1.1 ± 2.9 | -10.9 ± 4.2 | -0.7 ± 5.2 | -31.1 ± 8.1 |
| Total (GRACE) | No | -3.6 ± 1.5 | -15.7 ± 3.1 | -0.7 ± 2.9 | -9.4 ± 4.2 | -1.8 ± 5.2 | -31.2 ± 8.1 |
| SMB | Yes | -3.5 ± 0.4 | -8.2 ± 1.1 | -2.8 ± 0.2 | -12.1 ± 0.9 | 3.3 ± 0.4 | -23.3 ± 2.7 |
| SMB | No | -3.4 ± 0.4 | -8.5 ± 1.1 | -2.3 ± 0.2 | -14.6 ± 0.9 | 5.5 ± 0.4 | -23.3 ± 2.7 |
| -(Total-SMB) | Yes | -0.6 ± 1.6 | 7.4 ± 3.3 | -1.7 ± 2.9 | -1.2 ± 4.3 | 4.0 ± 5.2 | 7.8 ± 8.5 |
| -(Total-SMB) | No | 0.2 ± 1.6 | 7.2 ± 3.3 | -1.6 ± 2.9 | -5.2 ± 4.3 | 7.3 ± 5.2 | 7.9 ± 8.5 |
| Ice discharge | - | 0.5 ± 0.5 | 2.1 ± 0.7 | 0.2 ± 0.5 | -0.1 ± 0.4 | -0.1 ± 1.1 | 2.6 ± 1.5 |

to August.

Alexander et al. (2016) suggested that the inconsistency between the spatial resolution of GRACE-based estimates and that of the SMB model (11 km) may have a large impact onto the difference between the two time-series. In response to this con-
5  cern, we investigate the effect of using an alternative scheme to process SMB mass anomalies. Instead of processing them consistently with GRACE data, as it is explained in Sect. 2.2.3, we directly compute SMB-related mass anomalies per drainage system from the RACMO2.3 grid with a spatial resolution of 11 km. We find that the difference for entire Greenland is negligible: smaller than 2 Gt. For individual drainage systems, we find that the impact is also relatively small (<12 Gt) (see Fig. 5). Still, this effect shows a systematic behavior and may introduce some bias into GRACE-SMB estimates. For this reason, we
10  prefer to process SMB estimates consistently with GRACE data.




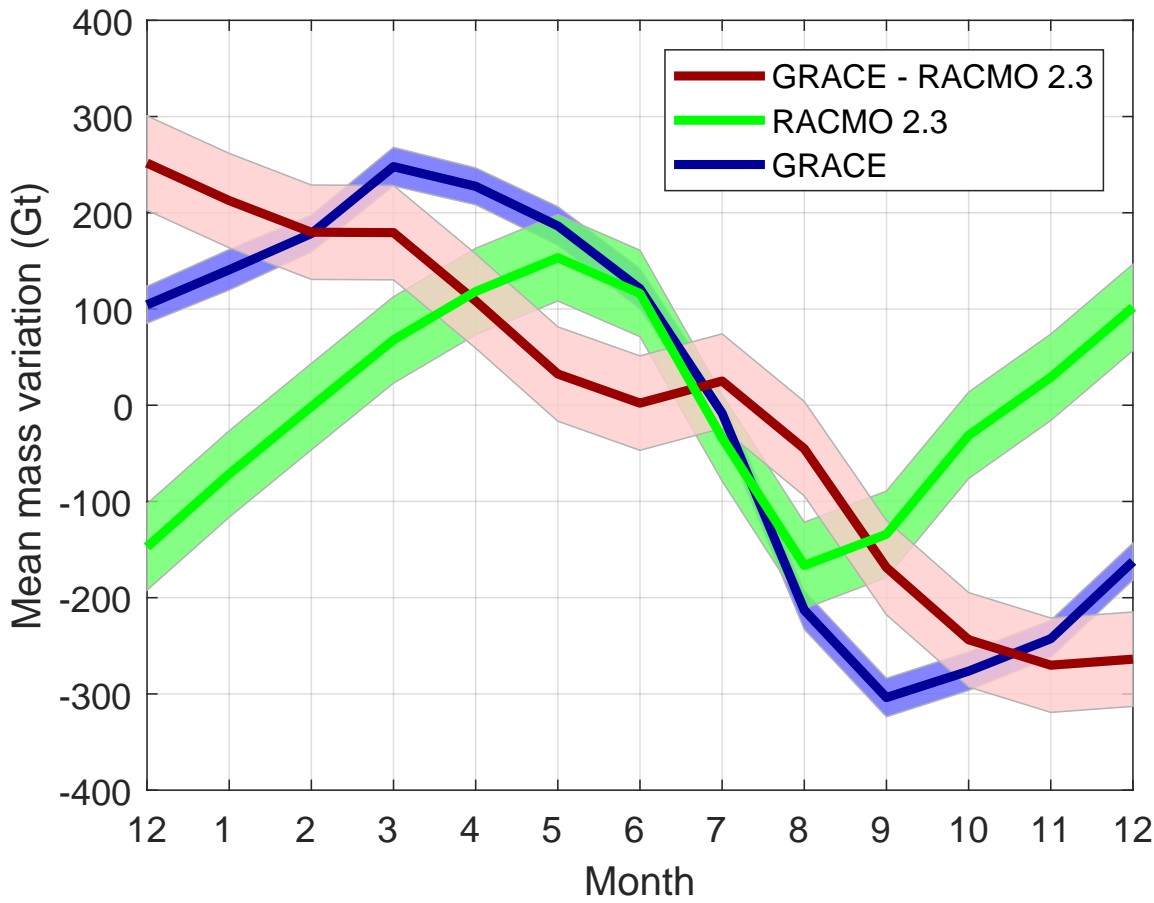

**Figure 4.** Greenland mean annual cycle of total mass anomalies from GRACE produced with data weighting (dark-blue), cumulative SMB anomalies (green) and the difference between them (brown) for the period 2003-2013. The latter curve reflects the cumulative sum of seasonal ice discharge variations and meltwater storage, as well as GRACE errors and SMB model bias. The shaded strips show the 1-$\sigma$ error bars. Labels at the horizontal axis indicate month of the year (Month 1 denotes January, month 12 is December).

### 3.2.1 Robustness of the Total-SMB residuals at the intra-annual time scale

The Total-SMB residuals show some periods of almost null variations (nearly flat segments in Fig. 4): February-March, May-July and November-December. The Total-SMB residuals represents the cumulative sum of ice discharge, meltwater storage, GRACE errors and SMB model biases. If we assume that the main contributor to the Total-SMB residuals is ice discharge, these

5  nearly-flat features would indicate that ice discharge is negligible or even negative, which is unphysical, since this implies that discharge is contributing to Greenland mass gain. Therefore, these features should be explained either by melt water retention



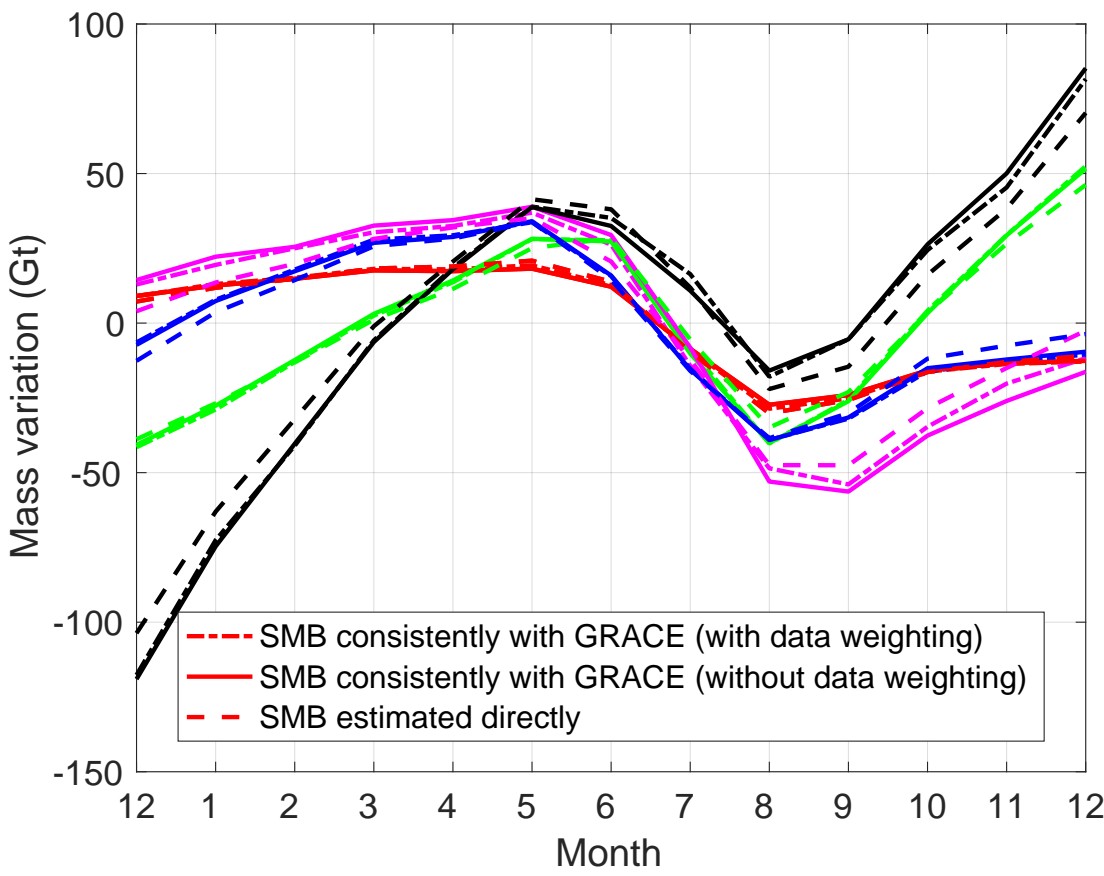

**Figure 5.** Simular to Fig. 4, but only for mean annual cycle of cumulative SMB anomalies of different drainage systems (SE: black, NW: green, NE: blue, N: red, SW: pink). The SMB outputs are computed in three different ways: (i) consistently with GRACE data (with or without data weighting); (ii) by a direct estimation of mass anomalies per drainage system.

or by errors in SMB- and GRACE-based estimates. In this section, we investigate the robustness of Total-SMB residuals with respect to those errors.

To assess a possible impact of errors in GRACE-based mass anomalies, we try different processing schemes in our variant of the mascon method. The following modifications of the GRACE data processing scheme were considered: i) retaining a

5 different number of eigenvalues of the noise covariance matrix $\mathbf{C_d}$ when inverting this matrix within the frame of the weighted least-squares estimation: 200, 400 or 600 eigenvalues (600 eigenvalues is the primary option); ii) different handling of the surrounding ocean: parameterization with one patch, parameterization with four patches (cf. Fig. A1), or without estimating mass anomalies over ocean (the latter is the primary option); iii) a different choice of spherical harmonic degree-one coefficients:



from Swenson et al. (2008), Cheng et al. (2013), or Sun et al. (2016) (the former is the primary option). Note that only one parameter varies at a time, while the primary option is chosen to define the other parameters (see also the Appendix). The optimal data weighting is exploited in all these experiments. In addition, we consider an effect of switching to the ordinary least-squares adjustment (when the data weighting is not used). To make the investigation even more comprehensive, we also

consider alternative GRACE-based estimates: from JPL mascon solutions by Watkins et al. (2015), CSR mascon solutions by Save et al. (2016), GSFC mascon solutions by Luthcke et al. (2013), and mascons solutions of Wouters et al. (2008).

The results are depicted in Figs. 6-7. Obviously, the presence and appearance of the nearly-zero Total-SMB month-to-month variations during February-March and November–December varies from case to case. For instance, when the surrounding

ocean is parameterized with four patches, the February-March feature becomes less flat; the November-December flat feature is not significant either in the estimates based on the ordinary least-squares estimator and on Wouters et al. (2008). In addition, the flat features of February–March and November–December do not appear in the Total-SMB residuals obtained from the CSR and JPL mascon solutions. Therefore, we infer that the nearly-zero Total-SMB variations during February-March and November-December which are clearly visible in Fig. 4, are likely caused by noise in the estimates. In the following, therefore,

they will not be discussed. On the other hand, the flat feature of May-July persists, no matter what processing parameters are chosen and which GRACE product is utilized. Therefore, it cannot be explained by uncertainties associated with GRACE data processing. Remarkably, switching data weighting on/off has the maximum impact on the obtained estimates of mass anomalies per calendar month. Since it is not clear at this moment which of the two options leads to better estimates, the results produced both without and with optimal data weighting will be considered in the further discussion.

To assess a possible impact of uncertainties in the SMB output, we analyze the SMB mass anomalies from RACMO2.3 and SNOWPACK. As shown in Fig. 8, the May–July flat feature persists in the Total-SMB residuals estimated from both RACMO2.3 and SNOWPACK. Therefore, we conclude that the May–July flat feature is likely not triggered by noise in the SMB estimates. As such, it must be attributed to a physical signal. We suggest that this signal is caused by temporary meltwater

storage.

### 3.2.2   A simple method to quantify temporary meltwater retention

According to RACMO2.3, meltwater is mostly produced between May and September, and peaks in July (cf. Fig. 9). Approximately 800 Gt of meltwater are produced on average in Greenland during the melt season from May to September, of which ∼250 Gt are estimated to refreeze within the snowpack, and the rest runs off of the ice sheet. RACMO2.3 does not calculate

lateral meltwater transport, i.e. the time lag between meltwater production and the moment when the runoff reaches the ocean. During late spring and early summer, this time is particularly long due to an inefficiency of the sub-glacial channelized network (Rennermalm et al., 2013) and replenishing of firn aquifers (mainly in the SE and NW) (Forster et al., 2014; Miège et al., 2016).

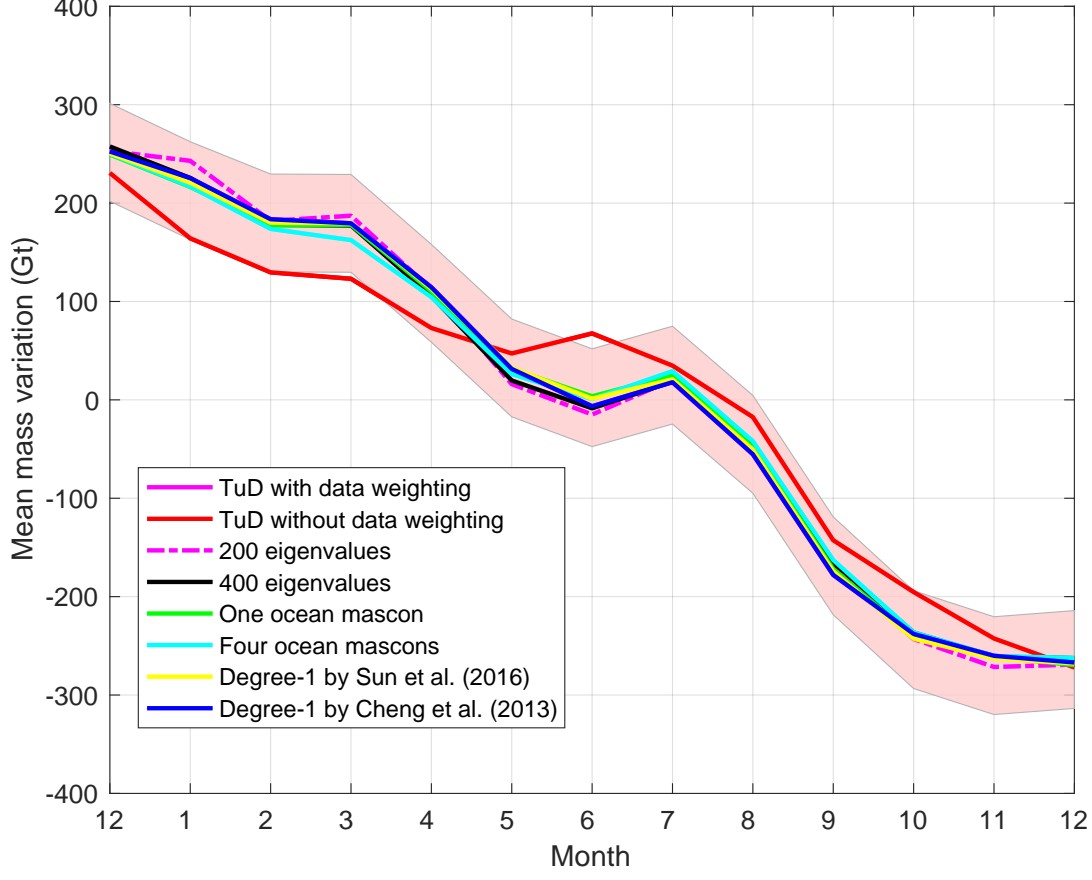

**Figure 6.** The mean mass anomalies per calendar month of the Total-SMB residuals over entire Greenland estimated by applying different processing shcemes.

In order to estimate the instantaneous amount of meltwater subject to runoff, we first fit the Total-SMB residuals in two periods, before and after the flat feature (i.e., in April-May and September-November), with a linear function. This function can be interpreted as an empirical estimation of the mean combined effect of ice discharge and the difference between the modelled meltwater refreezing and the actual one. Then, we force the mass budget at the beginning and the end of the melt season to be closed by subtracting the obtained linear function from the Total-SMB residuals (Fig. 10). In this way, we find that meltwater is retained in Greenland between May and October, with a $100 \pm 20$ Gt maximum in July. Note that the estimates of meltwater storage are sufficiently robust with respect to the choice of the GRACE-based mascon solution, but they may vary in a small range in timing and amplitude (Fig. 10).





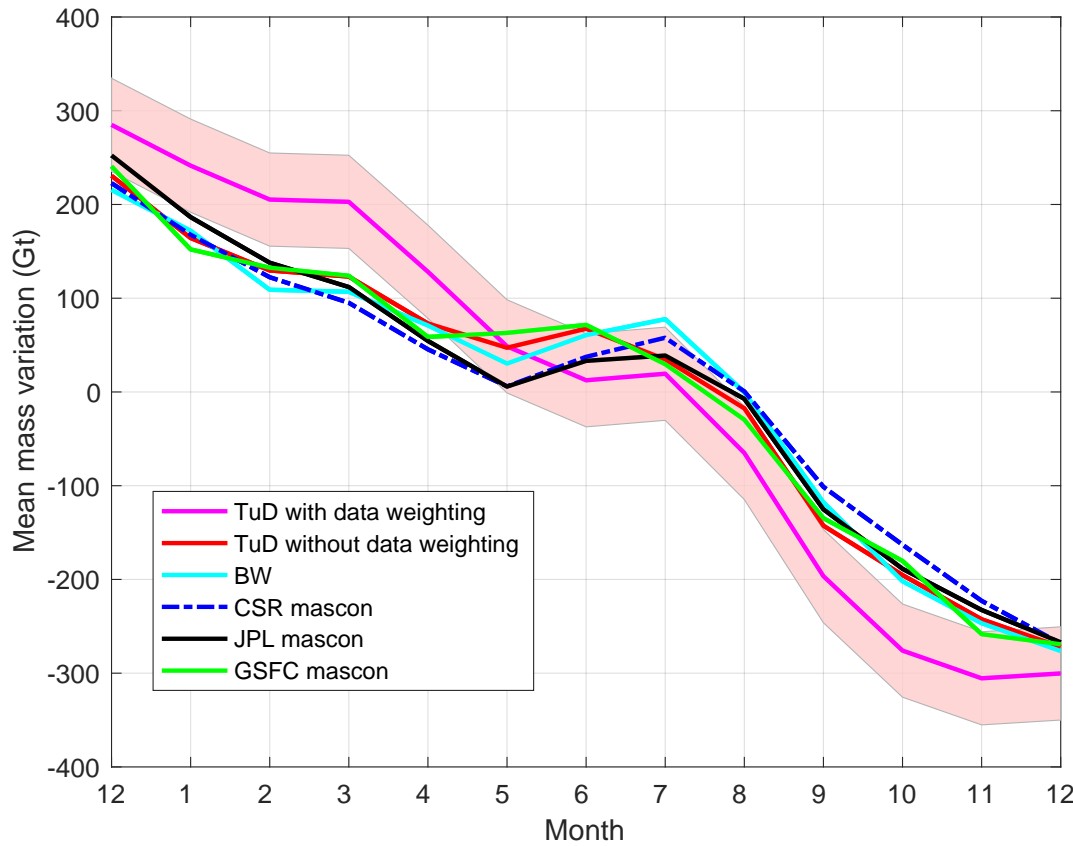

**Figure 7.** The mean mass anomalies per calendar month of the Total-SMB residuals over entire Greenland estimated from different GRACE solutions. "BW" refers to the solution of Wouters et al. (2008).

One may argue that the estimates of non-SMB mass anomalies per calendar month (our "Total-SMB" residuals) must reflect not only delayed meltwater discharge to the ocean, but also a variability of ice discharge. An effort to quantify the contribution of the latter is made here. To that end, we use an independent dataset of sub-annually resolved ice discharge estimates for 55 glaciers, which are mainly located in the NW and SE DSs, where the contribution of ice discharge to mass loss is the greatest. The sum of the obtained estimates over all 55 glaciers is shown in Fig. 11a. One can see that at the whole-ice-sheet scale, the increase in ice discharge during the melt season is minor in all years (∼10% or less). In the absence of complete coverage of the GrIS with observations of glacier velocities at the intra-annual time scale, we scale up the sum of ice discharge estimates by a factor of ∼2 to reach an agreement with the discharge over the entire GrIS in terms of the long-term linear trend (Enderlin et al., 2014). Similar to Fig. 10, we represent the ice discharge related mean mass anomaly per calendar month in terms of the



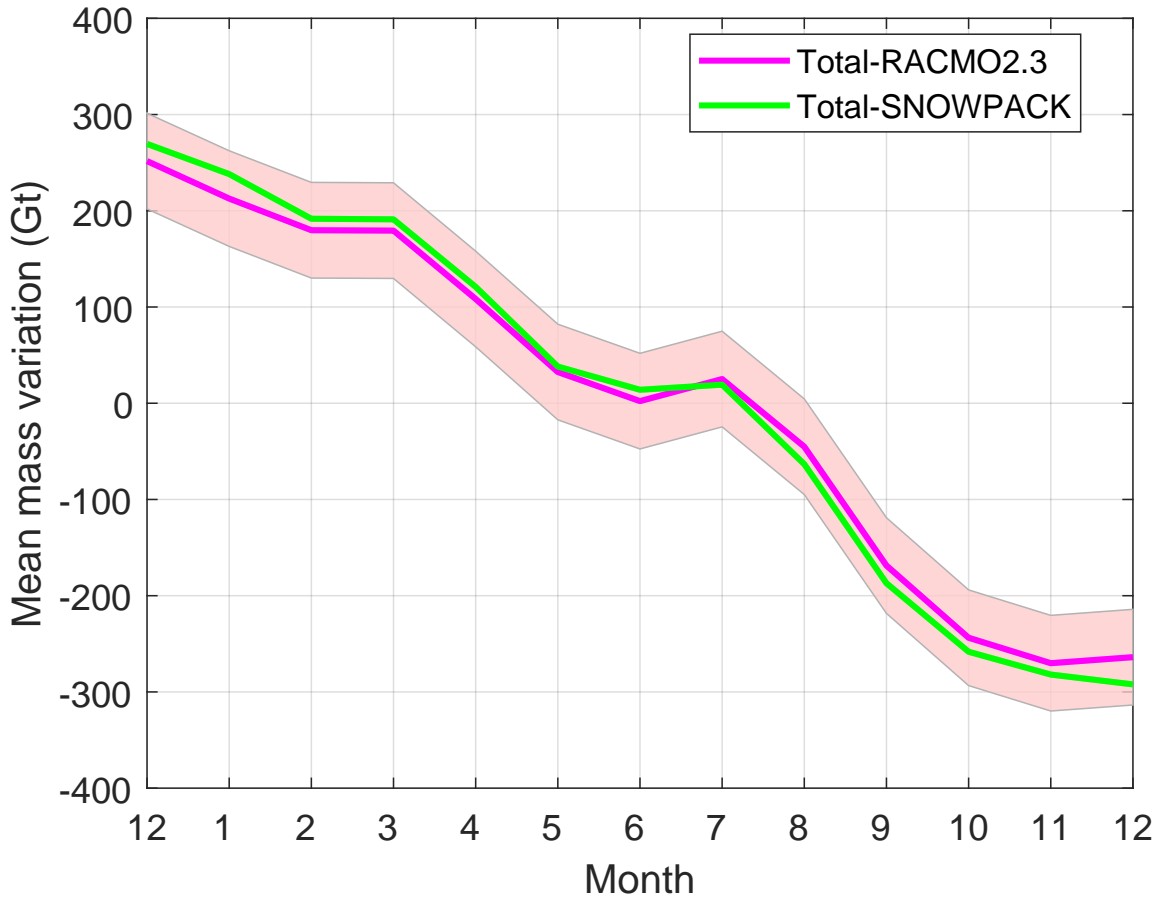

**Figure 8.** The mean mass anomalies per calendar month of the Total-SMB residuals over entire Greenland estimated from different SMB outputs.

deviation from the linear function fitting the values in April-May and September-November (cf. Fig. 12c). One can see that the effect of ice discharge amounts to only a few Gt, i.e. its contribution to the total signal is negligible. This supports our hypothesis that that delayed runoff is likely the major contributor to the signal isolated in Fig. 10.

5    Finally, we examine individual drainage systems (cf. Figs. 13-15). We refrain from an analysis of the SW and NE regions due to a relatively high level of noise in the obtained meltwater storage estimates. Regionally, the SE shows the largest meltwater accumulation per unit area. This is consistent with the fact that the rate of meltwater production is large in the sector (Fig. 9), as is the storage potential owing to high accumulation rates (Miège et al., 2016). In the N and NW regions, the signal related to



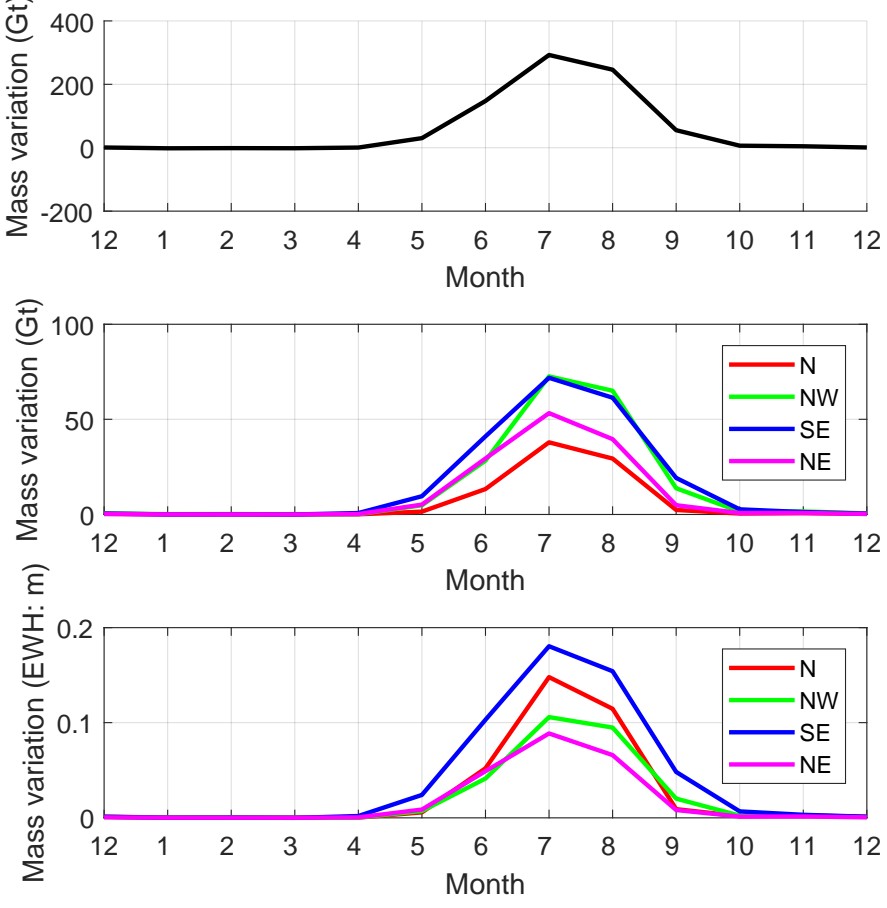

**Figure 9.** Mean monthly melt water production per calendar month (Gt) for whole Greenland (top), for individual drainage systems in Gt (middle), and for individual drainage systems in meters of Equivalent Water Height (bottom) modeled by RACMO 2.3.

meltwater storage is less pronounced, which can be explained by the dry climate of this region, meaning that less pore space is available in the firn layer to store liquid water.

In terms of the total mass, the largest meltwater accumulation takes place in the NW and SE regions: the contribution of each region may reach around 40 Gt in July-August (cf. Figs. 13-14).

5  As for the increase in ice discharge during the melt season, we find that it is relatively minor for both NW and SE DSs (less than 20% and 10%, respectively; see Fig. 11). As such, the contribution of ice discharge to the signal reported in Figs. 13-14 is minor for both DSs: not more than 2.0 and 0.3 Gt, respectively (cf. Fig. 12). Interestingly, a much larger increase in ice discharge during the melt season is found for the single major contributor to ice discharge, the Jakobshavn glacier: up to 60%

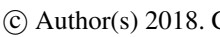



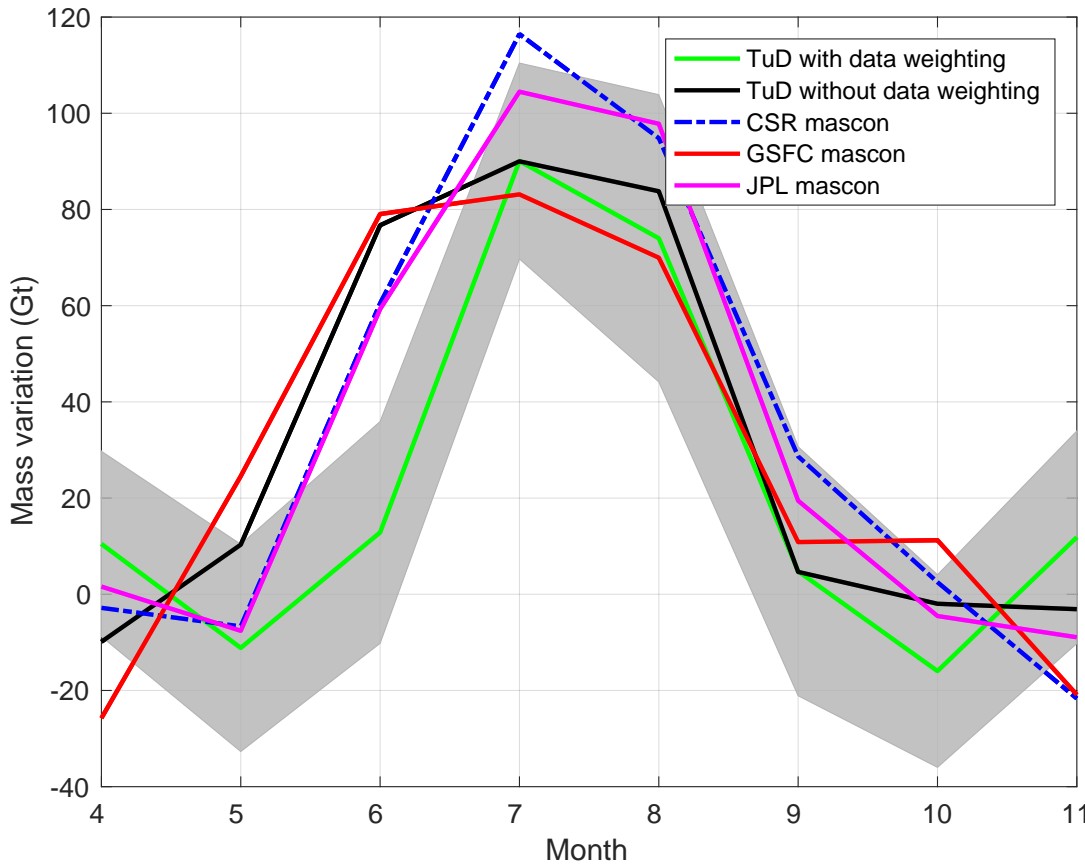

**Figure 10.** Estimates of seasonal meltwater storage, obtained as the monthly deviations from the April-May and September-November linear fit of "Total-SMB" residuals (brown line in Fig. 4): for the whole GrIS (in Gt). Labels along the horizontal axis represent months between April (4) and November (11). The shaded strip shows the 1-$\sigma$ error bar for the estimates by TuD with data weighting (the mean standard deviation is 23 Gt).

in 2012 (Fig. 16).

Note that the meltwater storage signal at the drainage system scale is present in all GRACE mascon solutions, but shows some discrepancies in the timing and amplitude. This means that further effort is still needed to improve the accuracy of

5  GRACE-based estimates.





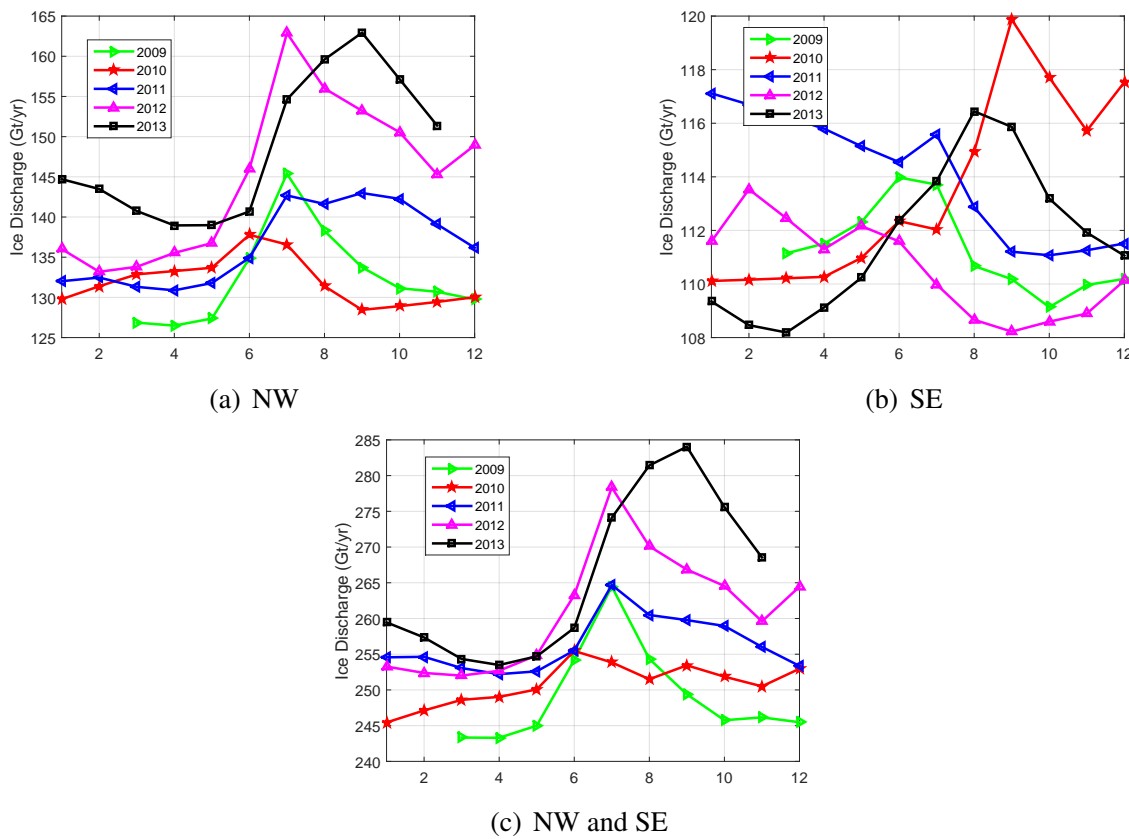

**Figure 11.** Monthly ice discharge estimates from 55 major marine-terminating glaciers for the glaciers in the NW DS (a) and the SE DS (b) individually; for the NW and SE drainage systems together (c). The unit is Gt/yr.

# 4 Conclusions

GRACE monthly solutions have been applied to systematically analyze the mass budget in the territory of Greenland at various temporal and spatial scales.



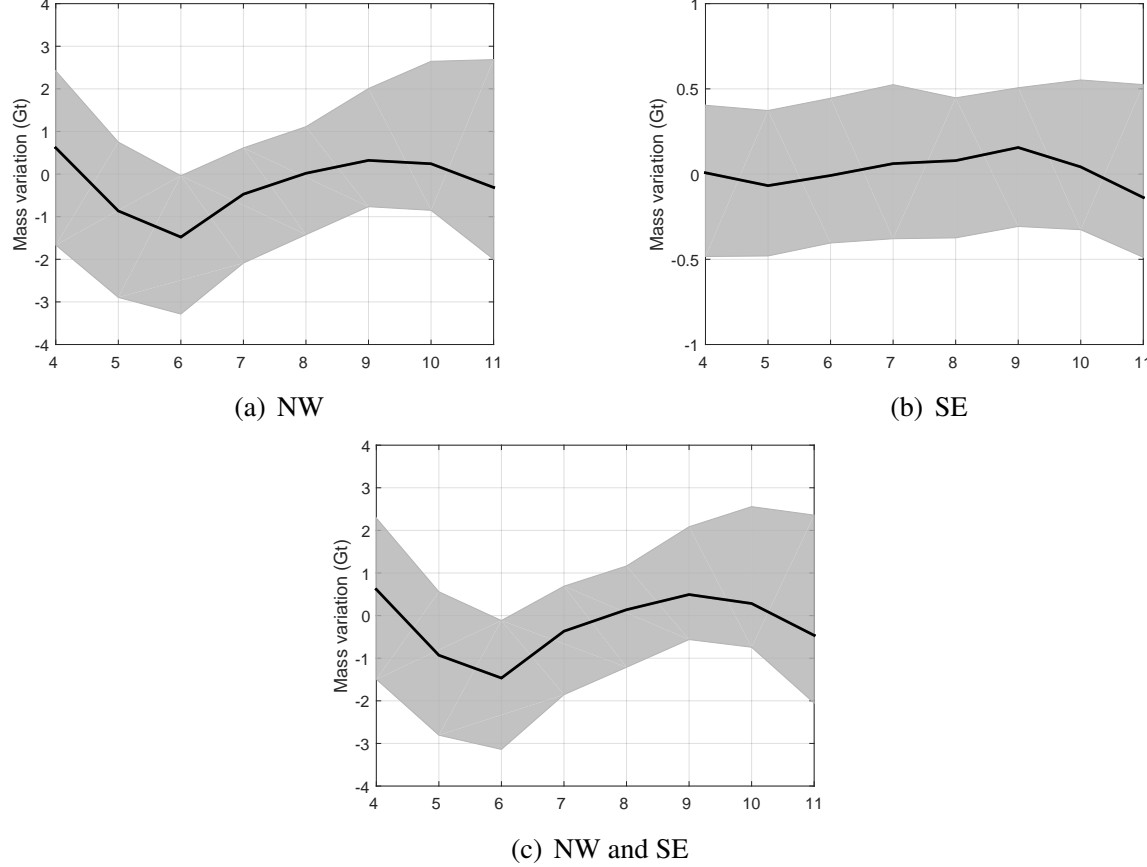

(a) NW

(b) SE

(c) NW and SE

**Figure 12.** Similar to Fig. 10, but for mean ice discharge related variations over 2009-2013. The cumulative mass anomalies are computed and upscaled to represent glaciers in NW (a), and in SE (b) drainage systems individually, and for glaciers in NW and SE together, upscaled to represent entire Greenland (c).

The obtained estimate of the mean rate of mass loss produced from CSR RL05 solutions with the new variant of mascon approach with and without data weighting is -277 ± 21 Gt/yr and -269 ± 21 Gt/yr over the period 2003-2012, respectively. The rate of SMB accumulation, as modelled by RACMO2.3, and processed consistently with GRACE data is 216 or 214 ± 122 Gt/yr, depending on whether data weighting is applied or not. The differences between GRACE- and RACMO-based trends with or without data weighting are 493 ± 124 Gt/yr or 483 ± 124 Gt/yr, which are consistent with 2003-2012 ice discharge observations by Enderlin et al. (2014): 520±31 Gt/yr. On the other hand, we observe relatively large discrepancies between the estimates for the SE and N DSs. Those discrepancies imply that the adopted climate model likely overestimates precipitation in the SE DS and underestimates it in the N DS.



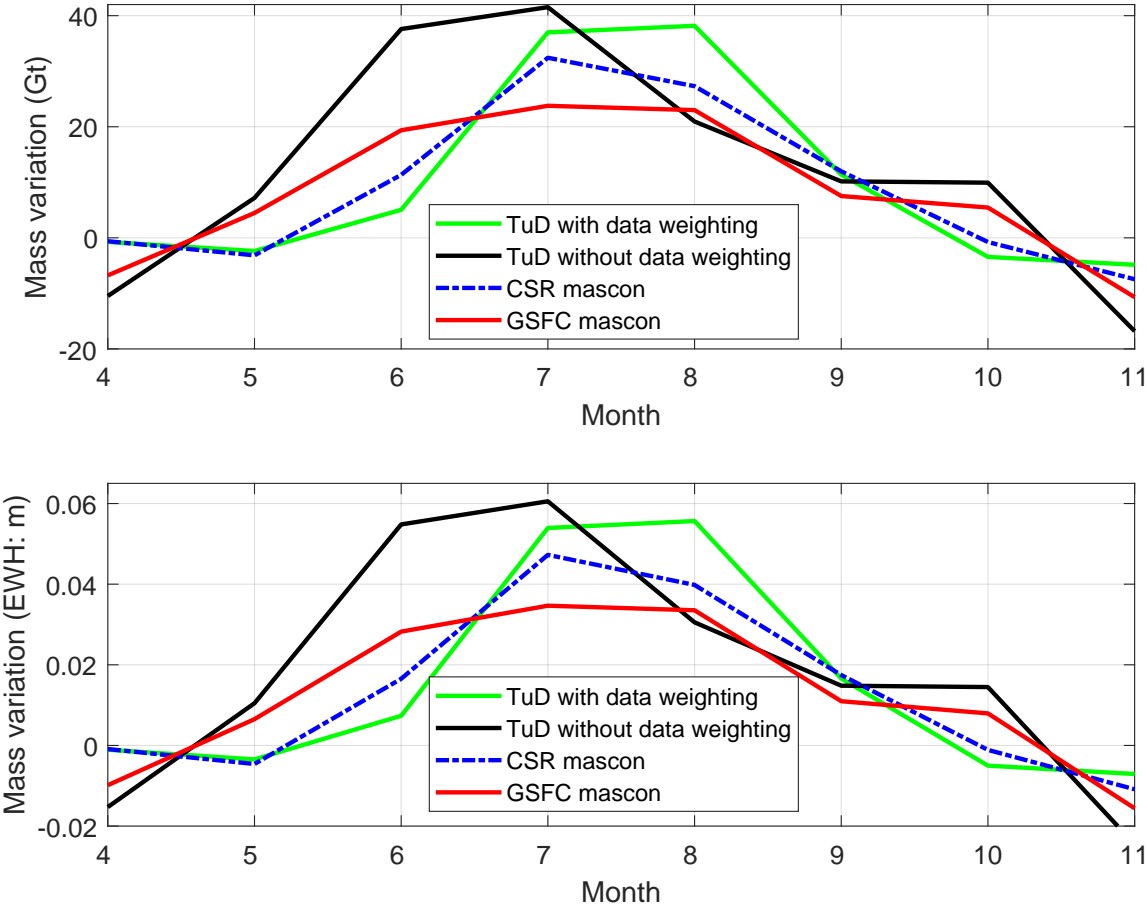

**Figure 13.** In line with Fig. 10, the top panel shows the estimates of seasonal meltwater storage in Gt for NW extracted from different mascon solutions. The mean standard deviations of the estimates is 25 Gt; it is not shown in the plot for the sake of its readability. The bottom panel, similar to (a), shows the estimates, but in meters of Equivalent Water Height. The mean standard deviations of the estimates is 0.04 m.

Our estimates of accelerations in SMB-related (-23.3±2.7 Gt/yr$^2$), ice discharge-related (2.6±1.5 Gt/yr$^2$), and total (-31.1±8.1 Gt/yr$^2$) mass anomalies are consistent within the error bar. Most of the observed acceleration is attributed to the SMB. This is consistent with Velicogna et al. (2014), who found that 79% of the mass loss acceleration can be explained by the contribution of SMB. Furthermore, our results indicate that most of the total mass acceleration observed by GRACE is
5    attributed to the SW and NW DSs, which is in agreement with Sasgen et al. (2010) and Velicogna et al. (2014).

We found a remarkable seasonal cycle in the difference between monthly total and SMB cumulative mass anomalies ("Total-SMB" residuals), which likely reflects significant meltwater storage in the early summer months due to an inefficiency of the





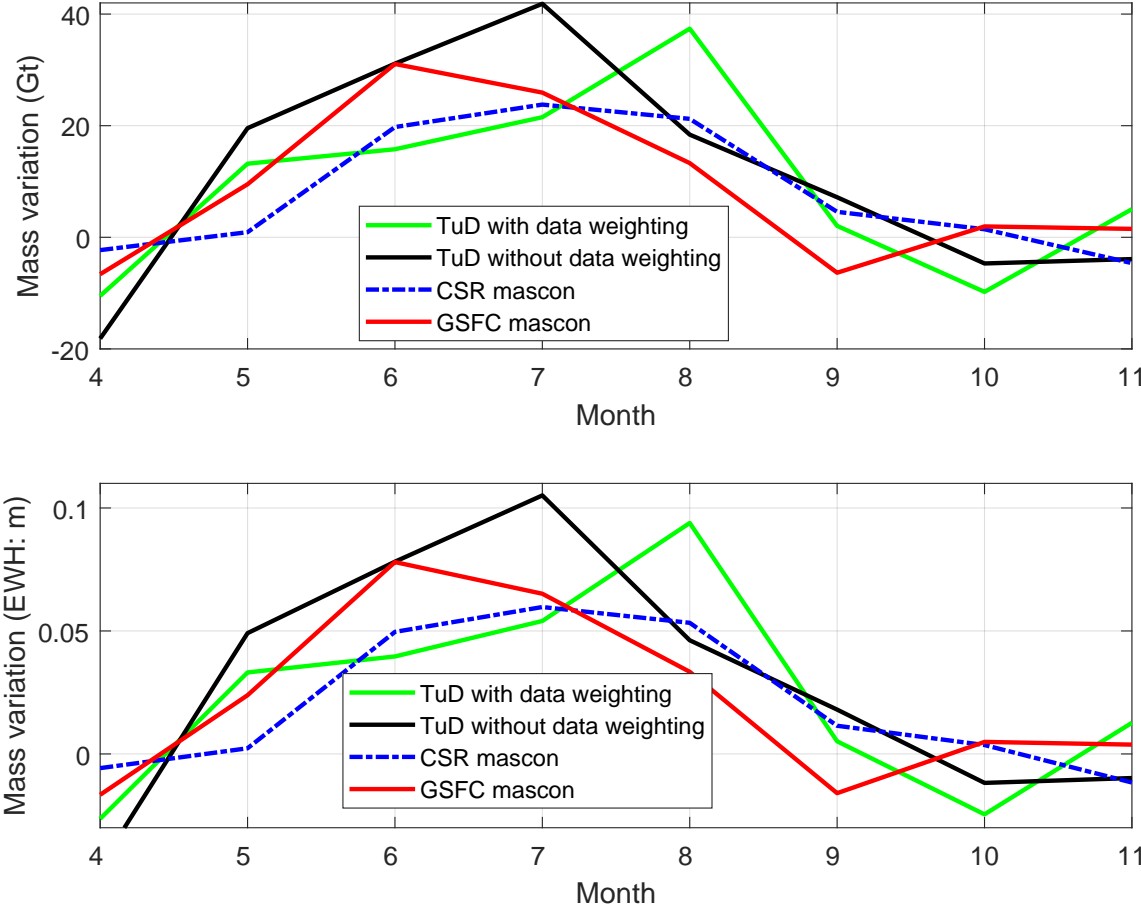

**Figure 14.** Similar to Fig. 13, but for SE. The mean standard deviations of the estimates in Gt and EWH are 26 Gt and 0.07 m, respectively.

sub-glacial channelized network. The maximum storage is observed in July: 80-120 Gt. To estimate the potential contribution of ice discharge to the observed signals, we exploited the estimates of ice discharge over 55 outlet glaciers obtained with the flux gate method. We showed that this contribution stays at the level of only a few Gt, i.e. plays a negligible role. We also analyzed the temporary meltwater storage per drainage system. Our results suggest that the meltwater storage is large in NW
5   and SE drainage systems, whereas it is weak in the northern DS.

A comparison of estimates derived from GRACE data with different processing parameters and from different mascon products (e.g., JPL, CSR, and GSFC) revealed a presence the temporary meltwater storage signal in all the considered solutions. At the same time, noticeable discrepancies are observed in timing and amplitude in meltwater storage estimates. These indicates

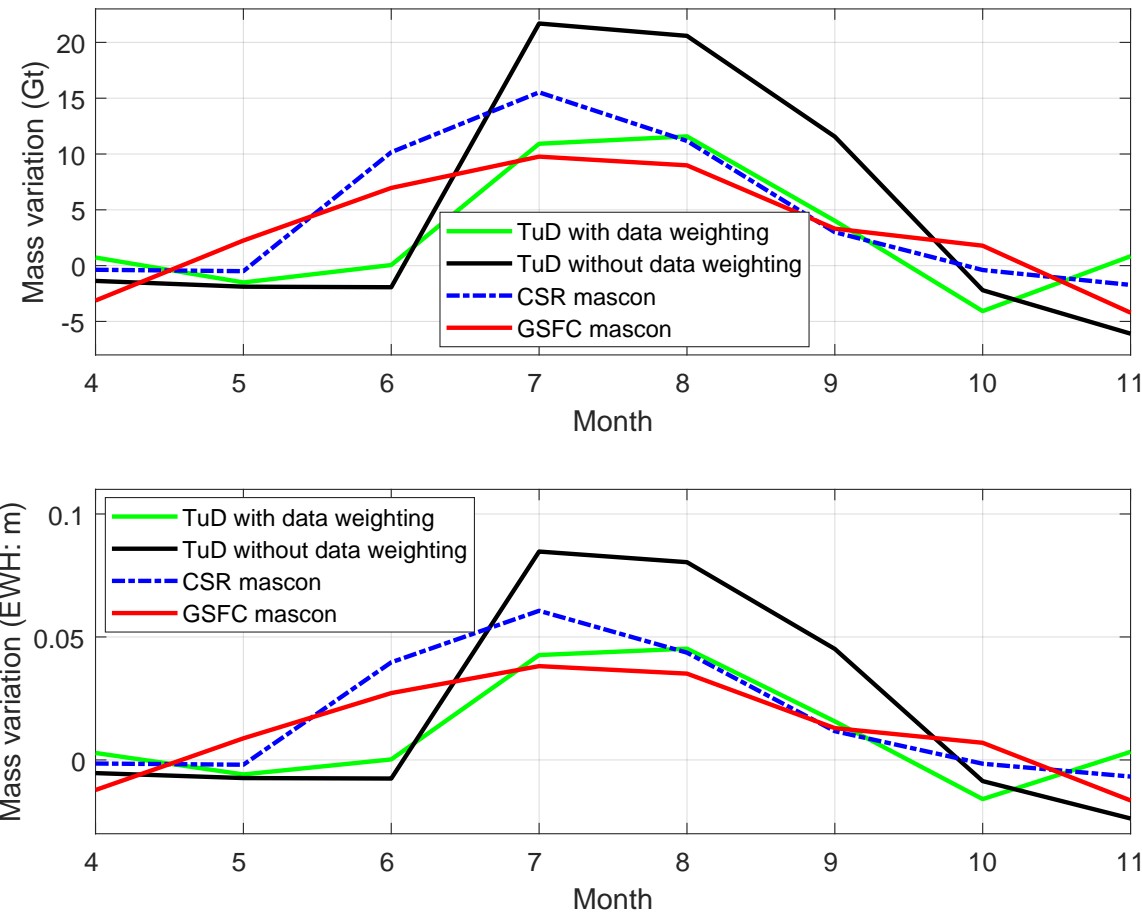

**Figure 15.** Similar to Fig. 13, but for N. The mean standard deviations of the estimates in Gt and EWH are 9 Gt and 0.03 m, respectively.

that further work is needed to improve GRACE-based estimates at both entire Greenland and drainage system scales.

Finally, this work illustrates the potential of combining multiple observational data sets and model output complemented by simple physical constraints, to better understand the contributors to GrIS mass variations at various time scales. Improving the estimates of (natural and forced) mass variations associated with individual processes is important for robust projections of future GrIS evolution and its contribution to sea level rise.





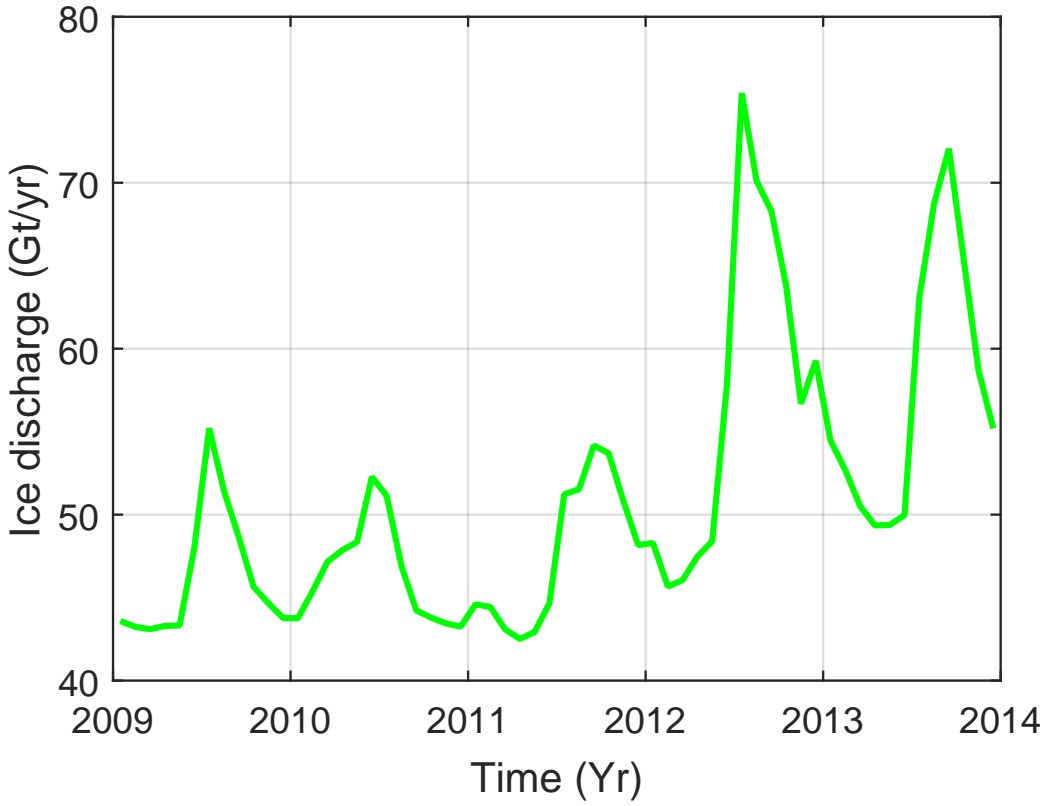

**Figure 16.** Monthly variations of ice discharge of Jakobshavn glacier over the period 2009-2013 (Gt/yr).

*Data availability.* GRACE Level-2 data and the corresponding error variance-covariance matrices used in this study is provided by the Center for Space Research at University of Texas at Austin. The mascon product estimated by the optimal data weighting scheme is available from the authors without conditions.

## 1   Appendix: Adopted method for estimating mass variation from GRACE

5   Our GRACE-based estimates of total mass variations are derived using a new variant of mascon approach (Ran, 2017). It can be considered as a further development of the computational procedure proposed by Forsberg and Reeh (2007) and Baur and Sneeuw (2011). The procedure consists of two steps. The goal of the first step is to synthesize temporal variations of gravity disturbances at a set of data points located at a specific satellite altitude (500 km). For brevity, they will be referred in the following as "gravity disturbances". The data points are homogeneously distributed over Greenland (extended with a 800-km buffer zone) with a 37.5-km separation. In parallel, the full error covariance matrix $\mathbf{C_d}$ of the gravity disturbances is computed on the basis of the full error covariance matrix of the spherical harmonic coefficients. In the second step, the synthesized gravity





disturbances are converted into mass anomalies per patch. The procedure is based on a linear functional model

$$\mathbf{d} = \mathbf{A}\mathbf{x} + \mathbf{n}, \tag{1}$$

where $\mathbf{d}$ is a vector composed of synthesized gravity disturbances, $\mathbf{x}$ is the vector consisting of mass anomalies, $\mathbf{A}$ is the design matrix relating the two vectors to each other in line with the Newton's attraction law, and $\mathbf{n}$ is the data noise vector. It is

worth mentioning that a straightforward implementation of this functional model may lead to some additional errors due to a spectral inconsistency between the matrix $\mathbf{A}$ and the data vector $\mathbf{d}$. The i-th column of $\mathbf{A}$ can be interpreted as a set of gravity disturbances caused by a unit mass anomaly in the i-th patch; its spatial spectrum is unlimited. On the other hand, the spatial spectrum of gravity disturbances $\mathbf{d}$ is limited to the maximum degree of the input spherical harmonic coefficients (in our case, 96). We eliminate this inconsistency by a low-pass filtering of all the columns of the matrix $\mathbf{A}$, so that the contribution of the

spherical harmonics above degree 96 is suppressed. The mass anomalies are computed from gravity disturbances by means of a least-squares adjustment

$$\mathbf{x} = (\mathbf{A}^{\mathbf{T}}\mathbf{P}\mathbf{A})^{-1}\mathbf{A}^{\mathbf{T}}\mathbf{P}\mathbf{d}, \tag{2}$$

where $\mathbf{P}$ is the weight matrix computed by an approximate inversion of the error covariance matrix of gravity disturbances $\mathbf{C_d}$. Note that the exact inverse of $\mathbf{C_d}$ cannot be computed because the matrix $\mathbf{C_d}$ is ill-posed. Therefore, an approximate inversion

of $\mathbf{C_d}$ is introduced, which is based on the eigenvalue decomposition of $\mathbf{C_d}$; only a limited number of the largest eigenvalues are retained. The usage of the matrix $\mathbf{P}$ ensures a (nearly) statistically optimal data weighting. From a preliminary numerical study we found that the optimal choice is to retain 600 eigenvalues. No regularization is applied in the course of inversion.

This procedure is used to produce one of the primary solutions, referred to as "solution obtained with data weighting"; the other primary solution, referred as "solution obtained without data weighting", is produced with the ordinary least-squares

adjustment

$$\mathbf{x} = (\mathbf{A}^{\mathbf{T}}\mathbf{A})^{-1}\mathbf{A}^{\mathbf{T}}\mathbf{d}. \tag{3}$$

*Author contributions.* P.D, R.K., and J.R. developed the methodology for GRACE data processing, J.R. processed the GRACE data; P.D., M.V., and J.R. interpreted the results based on GRACE data; M.V. initiated their comparison with ice discharge data; J.R, M.V, and P.D. wrote the manuscript, M.vdB., T.M., E.E., C.R.S., C.H.R. and B.W. provided additional data; B.W. contributed to the GRACE-intercomparison, all

authors commented on the manuscript.

*Competing interests.* The authors declare that they have no conflict of interest.

*Acknowledgements.* J. Ran thanks his sponsor, the Chinese Scholarship Council. M. Vizcaino is funded by the Dutch Technology Fellowship. M. van den Broeke and B. Wouters acknowledge funding from the Polar Program of the Netherlands Organization for Scientific Research



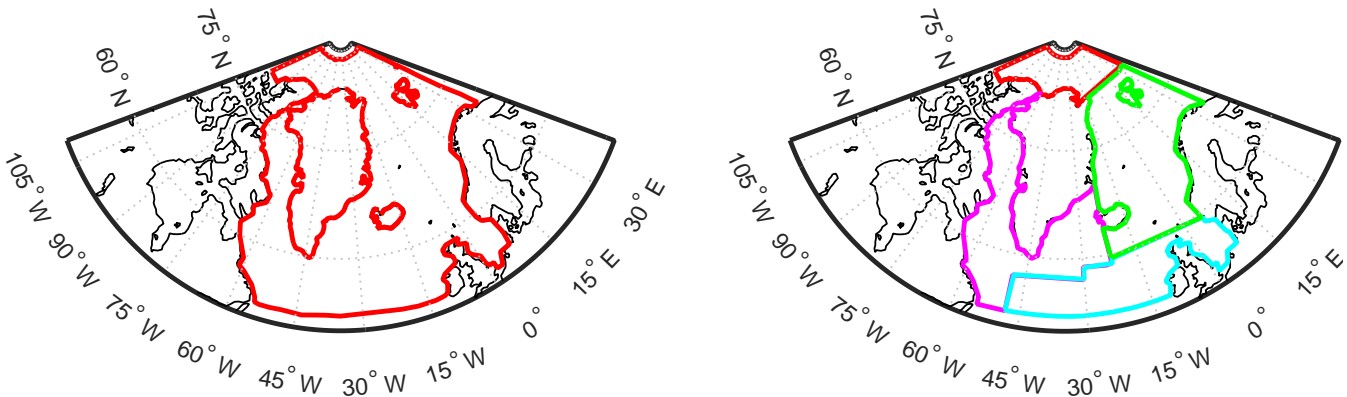

**Figure A1.** Parameterization of the ocean area around Greenland with one (left) and four (right) patches.

(NOW/NPP) and the Netherlands Earth System Science Center (NESSC). J. Ran has also been partly supported by the Strategic Priority
Research Program of the Chinese Academy of Sciences (XDB23030100) and the National Natural Science Foundation of China (41474063,
41674006 and 41674084).



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
