# Peer review of "Seasonal mass variations show timing and magnitude of meltwater storage in the Greenland ice sheet"

_The Cryosphere, 2018_

## Referee Comment (RC1) · Anonymous Referee #1 · 25 Apr 2018

**1   Overview**

Ran et al. (2018) quantify the amount of meltwater retained within the Greenland ice sheet using estimates of seasonal mass change from the Gravity Recovery and Climate Experiment (GRACE) and surface mass balance (SMB) outputs from regional climate models. The paper works to improve our understanding of mass changes over the Greenland Ice Sheet over seasonal time-scales. The work presented by the authors falls within the scope of The Cryosphere and is a very interesting approach to quantifying meltwater retainment. However, I am skeptical of the results and the estimated uncertainties. Uncertainty in the parameterization of meltwater refreezing and

retainment within a regional climate model could justify a relatively broad suite of results beyond the uncertainties presented here. I believe there are a number of issues that should be resolved before the publication of this manuscript.

**2 Broad comments**

- The overall organization and structure of the manuscript could be improved to make it to work better as a journal article.

- I think the weighting algorithm is penalizing semi-annual and interannual oscillations in your GRACE estimates. I'm not sure if it is causing signal leakage (to other mascons or out-of-the-system) or if it is simply dampening the signal. The semi-annual oscillation change could explain part of the discrepancy between your estimates that you mention on Page 14, Lines 17–18. This would affect the seasonal results for the weighted GRACE datasets. It might be better to simply use the unweighted model due to its correspondence with other independent GRACE estimates.

- Have you considered using the Fettweis et al. (2017) outputs of the MAR model? Estimates of regional snowmelt can be quite different compared with RACMO2.3. The model outputs are freely available online.

- I suggest detrending the data in the seasonal figures to be similar to Alexander et al. (2016). Alternatively, could remove the residual from an EMD method similar to (Luthcke et al., 2013) or a moving-average estimate. As is, a reader largely just sees the effects of the longterm trend in each estimate.

- Figures 13 to 15 could be merged in a single figure with subplots. I don't think the results need to be supplied in multiple units per region as it appears the averaging areas for each GRACE estimate are similar.
- The manuscript is not particularly well cited. Some of these procedures have been used before and some of these findings have been found before. A more thorough discussion about where this work fits in context of other studies would be beneficial.

**3 Line-by-line comments**

**Page 1, Lines 1–2:** The attribution portion of the first sentence is overly complex. I suggest removing "result of the changes in the complex ice-climate interactions that have been driven by global climate change"

**Page 1, Line 5:** Remove "Firstly, in agreement with previous estimates"

**Page 1, Lines 6–7:** I would mention the SMB-D estimate since you mention it is consistent with your GRACE results.

**Page 1, Lines 9:** The acceleration for SMB is mentioned here but not the acceleration in the GRACE ice mass estimate.

**Page 2, Lines 1–2:** I would suggest "The NASA/DLR Gravity Recovery and Climate Experiment (GRACE) mission is a powerful tool to monitor ice mass variations in Greenland, both the ice sheet (GrIS) and its peripheral glaciers."

**Page 2, Lines 2–6:** I suggest breaking up this sentence. "The total mass balance of the ice sheet represents the summation of processes in Equation 1: surface mass balance (SMB), ice discharge (ID) and en-glacial and sub-glacial mass variations $(\Delta m)$. GRACE ice mass balance is calculated after removing the impacts of Glacial Isostatic Adjustment (GIA), atmospheric and oceanic variability, and other time-variable geophysical processes."

**Page 2, Line 15:** I would remove "so that they have to be monitored on a regular basis"

**Page 2, Lines 16–17:** Spatial resolution of the monthly solutions is worse than the long-term trend?

**Page 2, Lines 21–23:** I would mention some regional studies of seasonal ice discharge or surface velocity change.

**Page 3, Line 5:** I suggest using utilized instead of exploited.

**Page 3, Lines 14–15:** Why not compute results for the total available GRACE-period (2002–2016 for the best quality data)? Could restructure your table to have periods of overlap with prior studies.

**Page 3, Lines 18–19:** If I understand correctly, geocenter variations are inherently zero in the native GRACE reference frame (center-of-mass) and not really missing.

**Page 3, Lines 24–25:** I would cite either Alexander et al. (2016) or Velicogna et al. (2014) for the GRACE-like processing of SMB data.

**Page 5, Line 26:** Why is relative italicized here?

**Page 5, Lines 26–27:** I would include that this assumes that the ice discharge and SMB were balanced during the reference period.

**Page 5, Lines 30–31:** Can you be more descriptive of what you mean by this sentence? Are you saying that SMB-D can never be positive or that GRACE-SMB shouldn't be negative? Also remove "For instance" from the beginning of the sentence.

**Page 6, Lines 1–3:** The inclusion of peripheral glaciers, ice caps and tundra regions has been investigated in other GRACE and SMB studies (Alexander et al., 2016; van Angelen et al., 2014).

**Page 6, Line 5:** I suggest "We estimate rates of ice discharge from two different datasets."

**Page 6, Line 13:** I would mention that you calculated monthly rates (assumed from Figure 11) of ice discharge and not simply intra-annual rates.

**Page 6, Lines 13–14:** Can you include a map with the locations of your 55 measured glaciers? For your sampled glaciers, what is the mixture between the 3 seasonal behaviors types described in Moon et al. (2014)? The overall seasonality in ice discharge will be dependent on this distribution.

**Page 7, Lines 10-11:** I would cite Jacob et al. (2012) as a reference for using 50% of the GIA signal as the estimated uncertainty. There are alternatives for calculating this number but the "back of the envelope" technique seems to be reasonable.

**Page 7, Lines 15–17:** I would write that you do not consider the uncertainty from atmospheric and ocean circulation as it is negligible. As written, it suggests Velicogna and Wahr (2013) were incorrect for calculating this error. Also, the seasonal and month-to-month uncertainty in these corrections is larger than the longterm.

**Page 9, Line 1:** As mentioned in the broad comments, I think the Ran et al. (2017) weighting algorithm is dampening some actual geophysical signal.

**Page 9, Lines 16-18:** What data suggests that precipitation is the component of SMB that causes the discrepancy for these two regions?

**Page 9, Lines 18-21:** This is suggesting that the ice discharge uncertainties from Enderlin et al. (2014) are underestimated for the region.

**Page 9, Lines 31–33:** Do the annual regressions include seasonal terms?

**Page 10, Figure 3:** Although the plots are relatively busy as is, it may be helpful to include the cumulative ice discharge change since you compare it with your GRACE-RACMO2.3 results.

**Page 11, Table 2:** 8 Gt/yr is a relatively large discrepancy between your weighted and unweighted results. Could you explain the difference? From Figure 2, the difference between the estimates doesn't seem that large over the longterm.

**Page 11, Line 8:** I would include percentages for the individual drainage systems (12 Gt could be relatively impactful).

**Page 11, Lines 9–10:** I would cite that the GRACE-like processing of SMB data is similar to Alexander et al. (2016) or Velicogna et al. (2014).

**Page 12, Figure 4:** The lag between GRACE and SMB that is shown here is quite different than the results in Alexander et al. (2016) and van Angelen et al. (2014). van Angelen et al. (2014) suggested the lag was 18 days, but here it is approximately 2 months. The results shown here suggests that the total mass begins changing 2 months before the onset of melt. Do you have a suggestion about what would cause this?

**Page 12, Line 1:** Minor comment: I suggest GRACE-SMB residuals versus Total-SMB residuals.

**Page 12, Line 3:** Minor comment: it would be GRACE errors and the correction uncertainties used in producing the ice mass estimates

**Page 13, Figure 5:** This should be probably be subplots as differentiating between 12 distinct lines is difficult.

**Pages 13–14:** While important for validating your results, I think the GRACE testing (different sets of eigenvalues, different processing centers and different geocenter estimates) should be moved into supplementary material.

**Page 14, Line 4:** Remove "To make the investigation even more comprehensive"

**Page 14, Line 8:** Remove "Obviously"

**Page 14, Line 9:** I'd go with "between GRACE estimates" versus "from case-to-case".

**Page 14, Line 8:** Remove "For instance"

**Page 14, Lines 17–18:** See the first broad comment about the weighting algorithm.

**Page 15, Lines 1–2:** You mention the amount of meltwater subject to runoff, but the SMB outputs should already include a portion of refreezing and retainment within the firn and snow layers. Would it be better for these results to add back the refreezing estimates to the SMB results to get the full en-glacial/sub-glacial retainment? A figure comparing the results with the modeled meltwater refreezing and retainment from the climate model could be beneficial.

**Page 16, Lines 1–2:** I suggest something like "Estimates of non-SMB mass anomalies could reflect the delayed release of meltwater into the ocean and the variability of ice discharge. We test the effects of ice discharge variability using a monthly-resolved dataset of ice discharge for 55 glaciers in Greenland. These glaciers are largely located in the NW and SE Greenland DS's, which are the largest contributors of ice mass wastage into the ocean." Then mention the number of different Moon et al. (2014) seasonal types within this dataset. As written, it seems that the variability in ice discharge is a negligible contributor to total mass seasonality, which might be too strong of language.

**Page 16, Lines 6–9:** Do you think the seasonality of glacier discharge would scale similarly to the mean fluxes?

**Page 18, Lines 6–7:** Should include error bars on these estimates. Uncertainty in both GRACE and SMB is large enough to justify a larger range than 0.3–2.0 Gt.

**Page 18, Lines 7–8:** Should cite Joughin et al. (2008) or Joughin et al. (2012) about the seasonality of Jakobshavn Isbræ. These large seasonal amplitudes are a relatively longterm observation.

**Page 19, Figure 10:** Is there a reason why the uncertainty estimate reduces from approximately 50 Gt for GRACE-SMB in previous plots to 23 Gt here?

**Page 22, Lines 1–2:** replace "mass anomalies are consistent within the error bar" to "mass anomalies are consistent within error bars".

**Page 22, Lines 2–3:** I suggest "Most of the observed acceleration in ice mass loss can be attributed to changes in SMB."

**Page 23, Line 3:** I suggest "Seasonality in ice discharge is on the order of a few Gt and is relatively negligible compared with meltwater retention."

**Page 25, Lines 5–6:** I suggest "The method is adapted from the computational procedures proposed by Forsberg and Reeh (2006) and Baur and Sneeuw (2011)."

**Page 25, Line 8:** Replace "The goal of the first step" with "1)"

**Page 25, Line 9:** Remove "For brevity, they will be referred in the following as "gravity disturbances"."

**Page 25, Line 10:** Remove "In parallel"

**Page 25, Line 11:** Replace "In the second step" with "2)"

**Page 26, Lines 4–5:** Remove "It is worth mentioning that"

**Page 26, Line 7:** Remove "On the other hand"

**Page 26, Lines 8:** Replace "in our case" with "here"

**Page 26, Line 14:** Remove "Note that"

**References**

P. M. Alexander, M. Tedesco, N.-J. Schlegel, S. B. Luthcke, X. Fettweis, and E. Larour. Greenland Ice Sheet seasonal and spatial mass variability from model simulations and GRACE (2003–2012). *The Cryosphere*, 10(3):1259–1277, June 2016. ISSN 1994-0424. doi: 10.5194/tc-10-1259-2016.

O. Baur and N. Sneeuw. Assessing Greenland ice mass loss by means of point-mass modeling: a viable methodology. *Journal of Geodesy*, 85(9):607–615, 2011. ISSN 1432-1394. doi: 10.1007/s00190-011-0463-1.

E. M. Enderlin, I. M. Howat, S. Jeong, M.-J. Noh, J. H. van Angelen, and M. R. van den Broeke. An improved mass budget for the Greenland ice sheet. *Geophysical Research Letters*, 41 (3):866–872, 2014. ISSN 1944-8007. doi: 10.1002/2013GL059010. 2013GL059010.

X. Fettweis, J. E. Box, C. Agosta, C. Amory, C. Kittel, C. Lang, D. van As, H. Machguth, and H. Gallée. Reconstructions of the 1900–2015 Greenland ice sheet surface mass balance using the regional climate MAR model. *The Cryosphere*, 11(2):1015–1033, 2017. doi: 10.5194/tc-11-1015-2017.

R. Forsberg and N. Reeh. Mass change of the Greenland Ice Sheet from GRACE. In *Gravity Field of the Earth – 1st meeting of the International Gravity Field Service*, volume 18, pages 454–458. Springer Verlag, 2006.

T. Jacob, J. Wahr, W. T. Pfeffer, and S. C. Swenson. Recent contributions of glaciers and ice caps to sea level rise. *Nature*, 482(7386):514–518, Feb. 2012. doi: 10.1038/nature10847.

I. R. Joughin, I. M. Howat, M. Fahnestock, B. Smith, W. Krabill, R. B. Alley, H. Stern, and M. Truffer. Continued evolution of Jakobshavn Isbrae following its rapid speedup. *Journal of Geophysical Research: Earth Surface*, 113(F4), 2008. ISSN 2156-2202. doi: 10.1029/2008JF001023. F04006.

I. R. Joughin, B. E. Smith, I. M. Howat, D. Floricioiu, R. B. Alley, M. Truffer, and M. A. Fahnestock. Seasonal to decadal scale variations in the surface velocity of Jakobshavn Isbrae, Greenland: Observation and model-based analysis. *Journal of Geophysical Research: Earth Surface*, 117(F2), May 2012. ISSN 2156-2202. doi: 10.1029/2011JF002110. F02030.

S. B. Luthcke, T. J. Sabaka, B. D. Loomis, A. A. Arendt, J. J. McCarthy, and J. Camp. Antarctica, Greenland and Gulf of Alaska land-ice evolution from an iterated GRACE global mascon solution. *Journal of Glaciology*, 59(216):613–631, Aug. 2013. ISSN 0022-1430. doi:
10.3189/2013JoG12J147.

T. Moon, I. R. Joughin, B. E. Smith, M. R. van den Broeke, W. J. van de Berg, B. Noël, and M. Usher. Distinct patterns of seasonal Greenland glacier velocity. *Geophysical Research Letters*, 41(20):7209–7216, Oct. 2014. ISSN 1944-8007. doi: 10.1002/2014GL061836. 2014GL061836.

J. Ran, P. Ditmar, R. Klees, and H. H. Farahani. Statistically optimal estimation of Greenland Ice Sheet mass variations from GRACE monthly solutions using an improved mascon approach. *Journal of Geodesy*, 2017. ISSN 1432-1394. doi: 10.1007/s00190-017-1063-5.

J. Ran, M. Vizcaino, P. Ditmar, M. R. van den Broeke, T. Moon, C. R. Steger, E. M. Enderlin, B. Wouters, B. Noël, C. H. Reijmer, R. Klees, and M. Zhong. Seasonal mass variations show timing and magnitude of meltwater storage in the Greenland ice sheet. *The Cryosphere Discussions*, 2018:1–30, 2018. doi: 10.5194/tc-2018-41.

J. H. van Angelen, M. R. van den Broeke, B. Wouters, and J. T. M. Lenaerts. Contemporary (1960–2012) Evolution of the Climate and Surface Mass Balance of the Greenland Ice Sheet. *Surveys in Geophysics*, 35(5):1155–1174, 2014. ISSN 1573-0956. doi: 10.1007/s10712-013-9261-z.

I. Velicogna and J. Wahr. Time-variable gravity observations of ice sheet mass balance: Precision and limitations of the GRACE satellite data. *Geophysical Research Letters*, 40(12): 3055–3063, 2013. ISSN 1944-8007. doi: 10.1002/grl.50527.

I. Velicogna, T. C. Sutterley, and M. R. van den Broeke. Regional acceleration in ice mass loss from Greenland and Antarctica using GRACE time-variable gravity data. *Geophysical Research Letters*, 41(22):8130–8137, 2014. ISSN 1944-8007. doi: 10.1002/2014GL061052.

---

## Referee Comment (RC2) · Anonymous Referee #2 · 27 Apr 2018

**GENERAL COMMENTS**

Although GRACE has been used extensively to monitor Greenland ice mass loss in the literature, the authors have carved out a nice little niche with this manuscript. They try to quantify summer meltwater retention in the ice sheet in terms of magnitude and timing. Overall this is a welcome addition to the ever widening list of cryospheric/oceanographic/hydrological processes and phenomena that can be revealed by combining GRACE results with appropriate models. It is understood that these are initial results that should be corroborated by further research. The authors allude to that more or less, when saying in section 3.2.1 that "these features should

be explained either by melt water retention or by errors in SMB- and GRACE-based estimates." Indeed, the "features" would easily fit inside the error bars. However, the following robustness/sensitivity analyses make it clear that the patterns are persisting. So, I would welcome to see this work published after certain revision, nevertheless.

SPECIFIC ISSUES

The abstract mentions three achievements: (1) obtaining mass loss estimates using their own methodology that are consistent with published mass estimates; (2) examining mass loss accelerations; and (3) quantifying meltwater storage. I find the first two points hardly relevant in view of the third point. Obtaining estimates that are consistent with what is known in the literature may be a good validation exercise to the authors, but hardly relevant for the reader. I am particularly suspicious of acceleration estimates given the relatively short time span of GRACE. Numerically one will always get some value and LS estimation and testing theory will tell you that this value is "significant". If, after successfull GRACE-FO launch and operation, we look at this part of the time series, say 20 years from now, we'd probably see a long-term signal that is decidedly different from parabolic behavior. Moreover, there is hardly any serious discussion of the acceleration in section 3.1.

I thus strongly recommend to remove or at least tone down the estimation aspects and the acceleration estimation. This recommendation includes the removal (from text and graphics) of anything to do with the non-weighted solutions. The difference between weighted and non-weighted solutions is a technical geodetic detail that might be reported elsewhere, but constitutes distraction here. I do believe that leaving all these aspects out will strengthen the main line of the manuscript.

I also recommend the authors to reconsider the use of the phrase "(surface) mass balance". To me it is a misnomer. Equations (1) or (2) are mass balances: a bookkeeping of inputs and outputs, sinks and sources, left sides and right sides. Individual terms should not be called "mass balance". I know that this terminology is by now ingrained

Interactive
comment

in cryospheric and GRACE communities, but I consider it wrong nevertheless. At a recent international conference I heard a presentation on "extreme mass balance". The author simply meant rapid ice mass loss.

At the same time, I have the feeling that the authors aren't clear about the mass balance equations themselves. Eqn (2) is a balance of fluxes, so SMB is a flux quantity too, say in units of Gt/yr. How about eqn (1)? If SMB and ID are flux quantities, then MB and delta-m should be, too. But delta-m is explained as a mass variation, i.e. time-variable mass (units of Gt), which is not a flux but a state quantity. And how about MB? And "melt water production" (fig. 9) sounds like a flux to me, although it is indicated in Gt units.

TECHNICAL DETAILS

- In the abstract mass losses are reported in terms of negative numbers. A negative mass loss is a mass increase.

- A hyphen is not a minus sign. Please write a minus sign, wherever it should be a minus sign, including in the legends and captions of graphs and in headers, etc.

- The acronym DS is not very helpful. Write out in full everywhere.

- Red pentagrams are not very visible in figure 1. Figure 1 can certainly be improved. Explain the blue patches outside Greenland briefly in the caption.

- Page 6, line 25: Least-squareS adjustment. (If it were singular, there is nothing to adjust).

- Be consistent in your mathematical typesetting. Take, e.g., the symbol "f" for flux gate in eqn (3) and in the line above. In the equation there are two different letters "f" and in the line above, the "f" should be set in math italic. Similarly, in line 24 (page 6), the v should be bold-math-italic. And check the N and i in the last line of page 6.

- Several graphs show mass variation with the unit (EWH: m). That is inappropriate.
The quantity is (expressed in) EWH and its unit is m.

- Page 26, Line 5: The sentence "No regularization is applied..." is preceded by an explanation of truncated eigenvalue decomposition. Now that is definitely regularization.

---

## Author Comment (AC1) · 29 Jul 2018

**Responses to the editor and reviewers**

We are grateful for the comments provided by the reviewers. Please find below our answers in red to the reviewers' comments in black and the suggested changes in the MS main text in red.

On behalf of all authors,

Jiangjun Ran

**Anonymous Referee #1**

1 Overview

Ran et al. (2018) quantify the amount of meltwater retained within the Greenland ice sheet using estimates of seasonal mass change from the Gravity Recovery and Climate Experiment (GRACE) and surface mass balance (SMB) outputs from regional climate models. The paper works to improve our understanding of mass changes over the Greenland Ice Sheet over seasonal time-scales. The work presented by the authors falls within the scope of The Cryosphere and is a very interesting approach to quantifying meltwater retainment. However, I am skeptical of the results and the estimated uncertainties. Uncertainty in the parameterization of meltwater refreezing and retainment within a regional climate model could justify a relatively broad suite of results beyond the uncertainties presented here. I believe there are a number of issues that should be resolved before the publication of this manuscript.

We have made significant changes to the manuscript to address the issues. In particular, we included SMB output from MAR 3.9, restructured the manuscript, toning down the part about the trend and acceleration, etc. For more details, please see the text below.

2 Broad comments
• The overall organization and structure of the manuscript could be improved to make it to work better as a journal article.

Done. In particular, we moved some parts of the manuscript to the appendix: i) estimation of trend and acceleration; ii) Total-SMB robustness test.

• I think the weighting algorithm is penalizing semi-annual and interannual oscillations in your GRACE estimates. I'm not sure if it is causing signal leakage (to other mascons or out-of-the-system) or if it is simply dampening the signal. The semi-annual oscillation change could explain part of the discrepancy between your estimates that you mention on Page 14, Lines 17–18. This would affect the seasonal results for the weighted GRACE datasets. It might be better to simply use the unweighted model due to its correspondence with other independent GRACE estimates.

It is, at this moment, unknown which one is closer to the reality. One could find that the suggestion is opposite to the one of another reviewer. Therefore, we prefer to keep both solutions and used them to assess uncertainty in our results.

In addition, we would like to add some comments on the methodology. First, the least-squares estimator is an unbiased estimator for positive definite weight matrices. Hence, the noticeable differences triggered by the usage of data weighting can only be explained by the impact of one particular noise realization that is present in actual data. Second, the inversion with data weighting is applied to GRACE monthly solutions individually. Therefore, semi-annual and interannual oscillations are not treated differently, as compared to other periods.

• Have you considered using the Fettweis et al. (2017) outputs of the MAR model? Estimates of regional snowmelt can be quite different compared with RACMO2.3. The model outputs are freely available online.

Done. We included MAR 3.9. The estimates of the meltwater storage are robust with respect to different SMB outputs.

• I suggest detrending the data in the seasonal figures to be similar to Alexander et al. (2016). Alternatively, could remove the residual from an EMD method similar to (Luthcke et al., 2013) or a moving-average estimate. As is, a reader largely just sees the effects of the longterm trend in each estimate.

Actually, at an early stage of this study, we worked with detrended data, i.e., the "relative" values. Later on, however, we decided to refrain from de-trending. This is because after detrending the data, we are more limited in our ability to interpret the results. For instance, we are unable to claim that a nearly flat/positive signal in the Total-SMB residuals is inconsistent with the properties of ice discharge. As far as the computation of meltwater storage is concerned, the estimates will not change because they already rely upon a variant of de-trending.

However, we could understand the concern of the reviewer (e.g., "a reader largely just sees the effects of the longterm trend in each estimate"). Therefore, we also show one more figure below where the Total-SMB estimates are detrended, and thereby one can see the anomalies do not interfere with the trend signal.

[Figure]

Figure S1 Detrended Total-SMB estimates. Different thin curves refer to different years. The thick black curve is the mean over all years (2003-2013).

• Figures 13 to 15 could be merged in a single figure with subplots. I don't think the results need to be supplied in multiple units per region as it appears the averaging areas for each GRACE estimate are similar.

We have merged Figs. 13-15 to be two figures with subplots (i.e., Fig. 9 and Fig. A6). It is worth to mention that the areas of the drainage systems (see Table 2) are quite different, and this has a direct impact on the shown estimates. For instance, NW (a relatively large DS) is one of the two largest contributors to the meltwater production in terms of total mass (units: Gt) (Fig. 5 middle), but by far not the largest one in terms of meters EWH (Fig. 5 bottom). Therefore, we suggest to keep the estimates in both Gt and meters EWH. But, in order to take the comments from the reviewer into account, we only include the estimates in Gt in the main text (Fig. 9). We move the estimates in meters EWH to the appendix (Fig. A6).

• The manuscript is not particularly well cited. Some of these procedures have been used before and some of these findings have been found before. A more thorough discussion about where this work fits in context of other studies would be beneficial.

We have included more references about the ice discharge of major outlet glaciers, for instance, Joughi et al. (2008, 2012). In addition, we also include a short discussion about the difference of the meltwater storage in this study and those in the literatures (Machguth et al. 2016, Harper et al. 2012, Forster et al. 2014, etc.). Jacob et al. (2012) is cited for the uncertainty of GIA model.

We add

"Furthermore, there are also meltwater storage estimates of Greenland ice sheet using in situ core data (e.g., Machguth et al. 2016, Harper et al. 2012, Forster et al. 2014). Usually, the authors collected the data during a short period, to characterize the state of the firn in a transect, to understand the capacity of the firn to store meltwater. These findings are then applied to the whole ice sheet based on a firn model. The meltwater storage estimated by those studies is significant, e.g., at the level of few hundreds or even thousand Gt. Those studies, however, are quite different, compared with this study. Those studies estimated the total meltwater storage in the firn, whereas in contrast, our study addresses (i) the water storage in all compartments (supra-, in-, and sub-glacial); (ii) at the short-time scale."

3 Line-by-line comments

Page 1, Lines 1–2: The attribution portion of the first sentence is overly complex. I suggest removing "result of the changes in the complex ice-climate interactions that have been driven by global climate change"

Done.

Page 1, Line 5: Remove "Firstly, in agreement with previous estimates"

Done.

Page 1, Lines 6–7: I would mention the SMB-D estimate since you mention it is consistent with your GRACE results.

Done. We included "(i.e., -304±126 Gt/yr)" in the main text.

Page 1, Lines 9: The acceleration for SMB is mentioned here but not the acceleration in the GRACE ice mass estimate.

We have removed the text about the acceleration in the abstract, based on the comment from the reviewer 2.

Page 2, Lines 1–2: I would suggest "The NASA/DLR Gravity Recovery and Climate Experiment (GRACE) mission is a powerful tool to monitor ice mass variations in Greenland, both the ice sheet (GrIS) and its peripheral glaciers."

Done.

We changed it to "The NASA/DLR Gravity Recovery and Climate Experiment (GRACE) mission is a powerful tool to monitor ice mass variations in Greenland, both the ice sheet (GrIS) and its peripheral glaciers and ice caps"

Page 2, Lines 2–6: I suggest breaking up this sentence. "The total mass balance of the ice sheet represents the summation of processes in Equation 1: surface mass balance (SMB), ice discharge (ID) and en-glacial

and sub-glacial mass variations (m). GRACE ice mass balance is calculated after removing the impacts of Glacial Isostatic Adjustment (GIA), atmospheric and oceanic variability, and other time-variable geophysical processes."

Done.

We changed it to "The total mass balance (TMB) of the ice sheet represents the summation of processes in Eq. 1: (i) surface mass balance (SMB), (ii) ice discharge (D), and (iii) mass variations ($\Delta$ m/($\Delta$ t$) which include all processes not related to SMB and ice discharge, for instance, en- and sub-glacial meltwater storage (MS). GRACE ice mass balance is calculated after removing the impacts of Glacial Isostatic Adjustment (GIA), atmospheric and oceanic variability, and other time-variable geophysical processes."

Page 2, Line 15: I would remove "so that they have to be monitored on a regular basis"

Done.

Page 2, Lines 16–17: Spatial resolution of the monthly solutions is worse than the longterm trend?

Yes. We believe so, since the random noise (i.e., the North-South oriented "stripe") could be largely reduced in the long-term trend.

Page 2, Lines 21–23: I would mention some regional studies of seasonal ice discharge or surface velocity change.

Done. We included some studies which discussed the seasonal ice velocities of limited numbers of marine terminating glaciers. We add in the main text as "The seasonal variabilities of the ice velocities of a few marine terminating glaciers were investigated by Howat et al. (2010), Ahlstrøm et al. (2013), Moon et al. (2015), etc."

Page 3, Line 5: I suggest using utilized instead of exploited.

Done.

Page 3, Lines 14–15: Why not compute results for the total available GRACE-period (2002–2016 for the best quality data)? Could restructure your table to have periods of overlap with prior studies.

This is because the limited data availability (from 2000 to 2012) for the ice discharge of 178 marine terminating glaciers in Greenland at the multi-years scale, though the GRACE data are available from 2002 to 2017. Thereby, we limit the time interval investigated in this study close to the maximal year (i.e., 2012) with ice discharge data available. In addition, for some estimates, e.g., the long-term trend, the time interval of 2003-2013 are also presented to be consistent with Velicogna et al, (2014) and other previous publications.

Page 3, Lines 18–19: If I understand correctly, geocenter variations are inherently zero in the native GRACE reference frame (center-of-mass) and not really missing.

Yes. We agree with the reviewer that since GRACE flied around the geocenter (center-of-mass, CM) of the Earth system, GRACE could not "see" variations in the degree one coefficients, which reflect the variations of the geocenter. Thereby, the degree-one coefficients are not included in the GRACE Level 2 product.
However, the estimates in the CM reference frame are not practical, since they mix up the effects of surface mass transport and the reaction of the entire Earth to that mass transport. This is the reason why the estimates in the Center-of-Figure (CF) reference frame are traditionally used. As soon as we introduce that reference frame, it is fair to say that degree-1 are missing in the officially provided GRACE solutions.

Page 3, Lines 24–25: I would cite either Alexander et al. (2016) or Velicogna et al. (2014) for the GRACE-like processing of SMB data.

Done. Even though there are some difference between the scheme by Ran et al. (2018) and by Alexander et al. (2016) or Velicogna et al. (2014), they indeed share the same basic idea of data post-processing of SMB, to be consistent with GRACE data. Thereby, in Sect. 2.2.4, we add
"This scheme is similar to the GRACE-like processing of SMB data by Alexander et al. (2016) and Velicogna et al. (2014)."

Page 5, Line 26: Why is relative italicized here?

This is because we wanted to stress that in previous studies, the cumulative SMB values were detrended, i.e., one removed the SMB trend over 1961-1990, by assuming that the ice sheet is at an equilibrium state. Therefore, those accumulative SMB values are relative values. However, in this study, we did not remove the SMB trend over 1961-1990.

Page 5, Lines 26–27: I would include that this assumes that the ice discharge and SMB were balanced during the reference period.

Done. We included the statement by the reviewer as "In other words, it assumes that the ice discharge and SMB were balanced during the reference period."

Page 5, Lines 30–31: Can you be more descriptive of what you mean by this sentence? Are you saying that SMB-D can never be positive or that GRACE-SMB shouldn't be negative? Also remove "For instance" from the beginning of the sentence.

We mean that GRACE-SMB should not be negative. We have changed the text as
"In this way, we are able to extract more information from the datasets: absolute mass anomalies related to ice discharge (i.e., the difference between GRACE- and SMB-based mass anomalies) cannot increase over time, which is a valuable constraint that facilitates the correct interpretation of the obtained results."

In addition, we have removed "for instance".

Page 6, Lines 1–3: The inclusion of peripheral glaciers, ice caps and tundra regions has been investigated in other GRACE and SMB studies (Alexander et al., 2016; van Angelen et al., 2014).

We have removed the text (Page 6, Lines 1-3).

Page 6, Line 5: I suggest "We estimate rates of ice discharge from two different datasets."

Done. We changed the text as "We examine ice discharge from two different datasets." Because we not only estimate the rates of ice discharge, but also investigated other quantities from the ice discharge datasets.

Page 6, Line 13: I would mention that you calculated monthly rates (assumed from Figure 11) of ice discharge and not simply intra-annual rates.

Changed as suggested. We changed the text from "intra-annual variations of ice discharge." to "monthly variations of ice discharge."

Page 6, Lines 13–14: Can you include a map with the locations of your 55 measured glaciers? For your sampled glaciers, what is the mixture between the 3 seasonal behaviors types described in Moon et al. (2014)? The overall seasonality in ice discharge will be dependent on this distribution.

The locations of 55 glaciers are shown as in Figure 1. We utilize the exactly the same dataset as Moon et al. (2014).

   We agree that the outlet glaciers show different seasonal behaviors types. Therefore, an extrapolation of the results based on 55 glaciers onto the entire GriS may be somewhat inaccurate if the behavior of the considered glaciers does not fully represent the behavior of the entire ice sheet. On the other hand, we found that the contribution of ice discharge variations to the observed Total-SMB residuals is negligible anyway. We believe, therefore, that even a relatively inaccurate estimation of the discharge variability is sufficient for our purposes.

Page 7, Lines 10-11: I would cite Jacob et al. (2012) as a reference for using 50% of the GIA signal as the estimated uncertainty. There are alternatives for calculating this number but the "back of the envelope" technique seems to be reasonable.

Done.

Page 7, Lines 15–17: I would write that you do not consider the uncertainty from atmospheric and ocean circulation as it is negligible. As written, it suggests Velicogna and Wahr (2013) were incorrect for calculating this error. Also, the seasonal and month-to-month uncertainty in these corrections is larger than the longterm.

Done.

Page 9, Line 1: As mentioned in the broad comments, I think the Ran et al. (2017) weighting algorithm is dampening some actual geophysical signal.

Please see our replies to the second broad comment.

Page 9, Lines 16-18: What data suggests that precipitation is the component of SMB that causes the discrepancy for these two regions?

This is one of the findings based on the analysis of results shown in this study and not taken from another dataset.

Page 9, Lines 18-21: This is suggesting that the ice discharge uncertainties from Enderlin et al. (2014) are underestimated for the region.

We have removed this sentence, because, currently, we are not sure whether the ice discharge from Enderlin et al. (2014) is underestimated for the region or not.

Page 9, Lines 31–33: Do the annual regressions include seasonal terms?

The seasonality is included by definition, as one unknown parameter per calendar month.

Page 10, Figure 3: Although the plots are relatively busy as is, it may be helpful to include the cumulative ice discharge change since you compare it with your GRACE-RACMO2.3 results.

Figure 3 is indeed very busy. By adding more curves, it will make the figure more unreadable. In the revised manuscript, we have toned down the parts of trend and acceleration estimates and moved parts of them (including Figure 3 in the previous version) to the appendix, as suggested by the second reviewer. However, in order to take the suggestion from the first reviewer into account, we include the cumulative ice discharge estimated by Enderlin et al. (2014) in Figure 2.

Page 11, Table 2: 8 Gt/yr is a relatively large discrepancy between your weighted and unweighted results. Could you explain the difference? From Figure 2, the difference between the estimates doesn't seem that large over the longterm.

This is because the data weighting scheme, which is designed to suppress the random noise, has some effect on the other error sources, e.g., the parameterization (model) error. More details are discussed in Ran et al. (2018a). The 8 Gt/yr difference is not clearly visible in the Figure 2, because it is relatively small as compared to the trend itself.

Ran, J., Ditmar, P., and Klees, R.: Optimal mascon geometry in estimating mass anomalies within Greenland from GRACE, Geophysical Journal International, p. ggy242, doi:10.1093/gji/ggy242, http://dx.doi.org/10.1093/gji/ggy242, 2018a.

Page 11, Line 8: I would include percentages for the individual drainage systems (12 Gt could be relatively impactful).

Done. We found that in term of percentage, it is smaller than 10%. We have included the largest percentage in the main text, as "… (<12 Gt, which is around 10% of the signal) (see Fig. 5). "
(see the largest discrepancy in SE in Dec (the most left side black curve in Fig. 5).)

Page 11, Lines 9–10: I would cite that the GRACE-like processing of SMB data is similar to Alexander et al. (2016) or Velicogna et al. (2014).

Done.

Page 12, Figure 4: The lag between GRACE and SMB that is shown here is quite different than the results in Alexander et al. (2016) and van Angelen et al. (2014). Van Angelen et al. (2014) suggested the lag was 18 days, but here it is approximately 2 months. The results shown here suggests that the total mass begins changing 2 months before the onset of melt. Do you have a suggestion about what would cause this?

Indeed, our finding is different from that of van Angelen et al. (2014). Both GRACE and RACMO show that an acceleration of mass losses starts in March. However, by comparing the timing of the peak values, this may be unfair because the presence of a large negative trend (due to discharge) in GRACE data may significantly change the timing of the peak. If one is interested to make a fairer comparison of the timing of peak values, we suggest to detrend both GRACE and SMB mass anomaly time-series before the calculation of mean mass anomalies per calendar month.

Page 12, Line 1: Minor comment: I suggest GRACE-SMB residuals versus Total-SMB residuals.

We agree that it is also possible to use GRACE-SMB. At the early stage of this study, we actually tried to use GRACE-SMB. But, finally, we chose to use Total-SMB, because we believe this could be more meaningful. The word "Total" refers to the physics behind the measured signal, whereas the word GRACE refers to a measurement instrument. Thus, the term "GRACE-SMB" suffers from an interval inconsistency. A consistent term is "GRACE-RACMO". However, a usage of that term would require an introduction of additional terms when alternative models are used to estimate the SMB.

Page 12, Line 3: Minor comment: it would be GRACE errors and the correction uncertainties used in producing the ice mass estimates

There is likely some misunderstanding. "GRACE errors" include also the "correction uncertainties", i.e. errors in the background models that are used in GRACE level-1b data processing, etc.

Page 13, Figure 5: This should be probably be subplots as differentiating between 12 distinct lines is difficult.

We agree that this figure is busy (currently it is Figure 4). On the other hand, we find the current representation very informative. It allows one to compare both different data processing strategies and the signals over different drainage systems. Furthermore, we believe that different curves can be distinguished in the plot with ease. We prefer, therefore, to keep the figure as it is.

Pages 13–14: While important for validating your results, I think the GRACE testing (different sets of eigenvalues, different processing centers and different geocenter estimates) should be moved into supplementary material.

Done.

Page 14, Line 4: Remove "To make the investigation even more comprehensive"

Done.

Page 14, Line 8: Remove "Obviously"

Done.

Page 14, Line 9: I'd go with "between GRACE estimates" versus "from case-to-case".

Done.

Page 14, Line 8: Remove "For instance"

Done.

Page 14, Lines 17–18: See the first broad comment about the weighting algorithm.

Please see the answer to the second question in the broad comments.

Page 15, Lines 1–2: You mention the amount of meltwater subject to runoff, but the SMB outputs should already include a portion of refreezing and retainment within the firn and snow layers. Would it be better for these results to add back the refreezing estimates to the SMB results to get the full en-glacial/sub-glacial retainment? A figure comparing the results with the modeled meltwater refreezing and retainment from the climate model could be beneficial.

Actually, we believe that Figure 9, which shows the total meltwater modelled by RACMO, is what the reviewer wants to see: runoff+ refreezing + retainment.

Page 16, Lines 1–2: I suggest something like "Estimates of non-SMB mass anomalies could reflect the delayed release of meltwater into the ocean and the variability of ice discharge. We test the effects of ice discharge variability using a monthly resolved dataset of ice discharge for 55 glaciers in Greenland. These glaciers are largely located in the NW and SE Greenland DS's, which are the largest contributors

of ice mass wastage into the ocean." Then mention the number of different Moon et al. (2014) seasonal types within this dataset. As written, it seems that the variability in ice discharge is a negligible contributor to total mass seasonality, which might be too strong of language.

Done.

Page 16, Lines 6–9: Do you think the seasonality of glacier discharge would scale similarly to the mean fluxes?

We assume that the seasonality of glacier discharge would scale similarly to the mean fluxes. But we do not know whether this assumption holds true in reality, because of a lack of data. In any case, we believe the contribution of ice discharge to the observed signal is minor.

Page 18, Lines 6–7: Should include error bars on these estimates. Uncertainty in both GRACE and SMB is large enough to justify a larger range than 0.3–2.0 Gt.

Done. We include the error bars for the estimates: "$2.0 \pm 1.9$ and $0.3 \pm 0.5$ Gt, respectively"

Page 18, Lines 7–8: Should cite Joughin et al. (2008) or Joughin et al. (2012) about the seasonality of Jakobshavn Isbræ. These large seasonal amplitudes are a relatively longterm observation.

Done.

Page 19, Figure 10: Is there a reason why the uncertainty estimate reduces from approximately 50 Gt for GRACE-SMB in previous plots to 23 Gt here?

The uncertainties in seasonal GRACE-SMB estimates in Fig. 3 and 10 (now Fig. 3 and 6) are computed as the root-sum-square of the standard deviations of noise in GRACE- and SMB-based estimates. For the noise in SMB estimates, it is assumed as 9% and 15% errors in modeled mean mass anomalies due to precipitation and runoff, respectively (van den Broeke et al. 2016).

However, the meltwater retention estimates shown in Figure 6, is computed after applying a linear regression to the GRACE-SMB residual estimates (as shown in Fig. 3). Therefore, when computing the noise in meltwater retention estimates related to SMB, we also compute it as 9% of the precipitation signal and 15% of the runoff signal, *after* applying the same linear regression to each component. Therefore, the noise in Fig. 6 becomes smaller than that in Fig. 3. In order to clarify our procedure to compute the noise in the meltwater retention estimates, we add
"The uncertainties of meltwater retention are computed as the root-sum-square of the standard deviations of noise in GRACE- and SMB-based estimates. It is worth to mention that, the error in the SMB estimates is then computed as assuming 9% and 15% errors in modeled mean mass anomalies due to precipitation and runoff signals, *after* applying the same linear regression function (see above) to each component, respectively.".

Page 22, Lines 1–2: replace "mass anomalies are consistent within the error bar" to "mass anomalies are consistent within error bars".

Done.

Page 22, Lines 2–3: I suggest "Most of the observed acceleration in ice mass loss can be attributed to changes in SMB."

Done.

Page 23, Line 3: I suggest "Seasonality in ice discharge is on the order of a few Gt and is relatively negligible compared with meltwater retention."

Done.

Page 25, Lines 5–6: I suggest "The method is adapted from the computational procedures proposed by Forsberg and Reeh (2006) and Baur and Sneeuw (2011)."

Done.

Page 25, Line 8: Replace "The goal of the first step" with "1)"

Done.

Page 25, Line 9: Remove "For brevity, they will be referred in the following as "gravity disturbances"."

Done.

Page 25, Line 10: Remove "In parallel"

Done.

Page 25, Line 11: Replace "In the second step" with "2)"

Done.

Page 26, Lines 4–5: Remove "It is worth mentioning that"

Done.

Page 26, Line 7: Remove "On the other hand"

Done.

Page 26, Lines 8: Replace "in our case" with "here"

Done.

Page 26, Line 14: Remove "Note that"

Done.

References

P. M. Alexander, M. Tedesco, N.-J. Schlegel, S. B. Luthcke, X. Fettweis, and E. Larour. Greenland Ice Sheet seasonal and spatial mass variability from model simulations and GRACE (2003–2012). The Cryosphere, 10(3):1259–1277, June 2016. ISSN 1994-0424. doi: 10.5194/tc-10-1259-2016.

O. Baur and N. Sneeuw. Assessing Greenland ice mass loss by means of point-mass modeling: a viable methodology. Journal of Geodesy, 85(9):607–615, 2011. ISSN 1432-1394. doi: 10.1007/s00190-011-0463-1.

E. M. Enderlin, I. M. Howat, S. Jeong, M.-J. Noh, J. H. van Angelen, and M. R. van den Broeke. An improved mass budget for the Greenland ice sheet. Geophysical Research Letters, 41 (3):866–872, 2014. ISSN 1944-8007. doi: 10.1002/2013GL059010. 2013GL059010.

X. Fettweis, J. E. Box, C. Agosta, C. Amory, C. Kittel, C. Lang, D. van As, H. Machguth, and H. Gallée. Reconstructions of the 1900–2015 Greenland ice sheet surface mass balance using the regional climate MAR model. The Cryosphere, 11(2):1015–1033, 2017. doi: 10.5194/tc-11-1015-2017.

R. Forsberg and N. Reeh. Mass change of the Greenland Ice Sheet from GRACE. In Gravity Field of the Earth – 1st meeting of the International Gravity Field Service, volume 18, pages 454–458. Springer Verlag, 2006.

T. Jacob, J. Wahr, W. T. Pfeffer, and S. C. Swenson. Recent contributions of glaciers and ice caps to sea level rise. Nature, 482(7386):514–518, Feb. 2012. doi:10.1038/nature10847.

I. R. Joughin, I. M. Howat, M. Fahnestock, B. Smith, W. Krabill, R. B. Alley, H. Stern, and M. Truffer. Continued evolution of Jakobshavn Isbrae following its rapid speedup. Journal of Geophysical Research: Earth Surface, 113(F4), 2008. ISSN 2156-2202. doi:10.1029/2008JF001023. F04006.

I. R. Joughin, B. E. Smith, I. M. Howat, D. Floricioiu, R. B. Alley, M. Truffer, and M. A. Fahnestock. Seasonal to decadal scale variations in the surface velocity of Jakobshavn Isbrae, Greenland: Observation and model-based analysis. Journal of Geophysical Research: Earth Surface, 117(F2), May 2012. ISSN 2156-2202. doi: 10.1029/2011JF002110. F02030.

S. B. Luthcke, T. J. Sabaka, B. D. Loomis, A. A. Arendt, J. J. McCarthy, and J. Camp. Antarctica, Greenland and Gulf of Alaska land-ice evolution from an iterated GRACE global mascon solution. Journal of Glaciology, 59(216):613–631, Aug. 2013. ISSN 0022-1430. doi: 10.3189/2013JoG12J147.

T. Moon, I. R. Joughin, B. E. Smith, M. R. van den Broeke, W. J. van de Berg, B. Noël, and M. Usher. Distinct patterns of seasonal Greenland glacier velocity. Geophysical Research Letters, 41(20):7209–7216, Oct. 2014. ISSN 1944-8007. doi: 10.1002/2014GL061836. 2014GL061836.

J. Ran, P. Ditmar, R. Klees, and H. H. Farahani. Statistically optimal estimation of Greenland Ice Sheet mass variations from GRACE monthly solutions using an improved mascon approach. Journal of Geodesy, 2017. ISSN 1432-1394. doi: 10.1007/s00190-017-1063-5.

J. Ran, M. Vizcaino, P. Ditmar, M. R. van den Broeke, T. Moon, C. R. Steger, E. M. Enderlin, B. Wouters, B. Noël, C. H. Reijmer, R. Klees, and M. Zhong. Seasonal mass variations show timing and magnitude of meltwater storage in the Greenland ice sheet. The Cryosphere Discussions, 2018:1–30, 2018. doi: 10.5194/tc-2018-41.

J. H. van Angelen, M. R. van den Broeke, B. Wouters, and J. T. M. Lenaerts. Contemporary (1960–2012) Evolution of the Climate and Surface Mass Balance of the Greenland Ice Sheet. Surveys in Geophysics, 35(5):1155–1174, 2014. ISSN 1573-0956. doi:10.1007/s10712-013-9261-z.

I. Velicogna and J. Wahr. Time-variable gravity observations of ice sheet mass balance: Precision and limitations of the GRACE satellite data. Geophysical Research Letters, 40(12): 3055–3063, 2013. ISSN 1944-8007. doi: 10.1002/grl.50527.

I. Velicogna, T. C. Sutterley, and M. R. van den Broeke. Regional acceleration in ice mass loss from Greenland and Antarctica using GRACE time-variable gravity data. Geophysical Research Letters, 41(22):8130–8137, 2014. ISSN 1944-8007. doi: 10.1002/2014GL061052.

---

## Author Comment (AC3) · 29 Jul 2018

**Responses to the editor and reviewers**

We are grateful for the comments provided by the reviewers. Please find below our answers in red to the reviewers' comments in black and the suggested changes in the MS main text in red.

On behalf of all authors,

Jiangjun Ran

**Anonymous Referee #2**

GENERAL COMMENTS

Although GRACE has been used extensively to monitor Greenland ice mass loss in the literature, the authors have carved out a nice little niche with this manuscript. They try to quantify summer meltwater retention in the ice sheet in terms of magnitude and timing. Overall this is a welcome addition to the ever widening list of cryospheric/oceanographic/hydrological processes and phenomena that can be revealed by combining GRACE results with appropriate models. It is understood that these are initial results that should be corroborated by further research. The authors allude to that more or less, when saying in section 3.2.1 that "these features should be explained either by melt water retention or by errors in SMB- and GRACE-based estimates." Indeed, the "features" would easily fit inside the error bars. However, the following robustness/sensitivity analyses make it clear that the patterns are persisting. So, I would welcome to see this work published after certain revision, nevertheless.

We are grateful for the comments from the reviewer. We also agree with the reviewer that this study presents a novel application of GRACE, and is an initial study. Future studies are necessary to investigate the issue of meltwater storage, and make it clearer.

SPECIFIC ISSUES

The abstract mentions three achievements: (1) obtaining mass loss estimates using their own methodology that are consistent with published mass estimates; (2) examining mass loss accelerations; and (3) quantifying meltwater storage. I find the first two points hardly relevant in view of the third point. Obtaining estimates that are consistent with what is known in the literature may be a good validation exercise to the authors, but hardly relevant for the reader. I am particularly suspicious of acceleration estimates given the relatively short time span of GRACE. Numerically one will always get some value and LS estimation and testing theory will tell you that this value is "significant". If, after successfull GRACE-FO launch and operation, we look at this part of the time series, say 20 years from now, we'd probably see a long-term signal that is decidedly different from parabolic behavior. Moreover, there is hardly any serious discussion of the acceleration in section 3.1.

I thus strongly recommend to remove or at least tone down the estimation aspects and the acceleration estimation.

Done. We have tried to tone down the parts of the long-term trend and acceleration, by moving a large part of text and figures related to trend and acceleration to be an appendix. In addition, we removed the second achievements (as indicated as (2) by the reviewer) about the accelerations from the abstract and the conclusion section. In addition, we also agree that the acceleration estimates cannot be blindly extrapolated onto a longer time interval and may not represent properly the ice sheet behavior at the decadal time scale, because of the large climate variability in a limited time span of data. We included a short discussion in the main text to discuss this issue, based on Wouters et al. (2013).

This recommendation includes the removal (from text and graphics) of anything to do with the non-weighted solutions. The difference between weighted and non-weighted solutions is a technical geodetic detail that might be reported elsewhere, but constitutes distraction here. I do believe that leaving all these aspects out will strengthen the main line of the manuscript.

We agree with the reviewer that the investigation of the difference of weighted and non-weighted solutions is a technical geodetic detail, and may cause distraction. However, the first reviewer asked the opposite to remove the weighted one. Actually, at this moment, we do not know which one (weighted or unweighted) leads to better quality. Therefore, we decide to keep both of them, in order to inform the readers.

I also recommend the authors to reconsider the use of the phrase "(surface) mass balance". To me it is a misnomer. Equations (1) or (2) are mass balances: a bookkeeping of inputs and outputs, sinks and sources, left sides and right sides. Individual terms should not be called "mass balance". I know that this terminology is by now ingrained in cryospheric and GRACE communities, but I consider it wrong nevertheless. At a recent international conference I heard a presentation on "extreme mass balance". The author simply meant rapid ice mass loss.

MB is sometimes called Total Mass Balance or Ice Sheet Mass Balance, and SMB is not strictly the surface mass balance but the climatic mass balance, because it incorporates sub-surface processes such as refreezing and retention. We understand and respect that there are different terminologies (see P6 in <GLOSSARY OF GLACIER MASS BALANCE AND RELATED TERMS>).

In this study, we follow the standard terminology (see < GLOSSARY OF GLACIER MASS BALANCE AND RELATED TERMS > by International Association of Cryospheric Sciences (IACS) in 2011). We have adjusted the text and Eqs. (1) and (2) accordingly.

At the same time, I have the feeling that the authors aren't clear about the mass balance equations themselves. Eqn (2) is a balance of fluxes, so SMB is a flux quantity too, say in units of Gt/yr. How about eqn (1)? If SMB and ID are flux quantities, then MB and delta-m should be, too. But delta-m is explained as a mass variation, i.e. time-variable mass (units of Gt), which is not a flux but a state quantity. And how about MB? And "melt water production" (fig. 9) sounds like a flux to me, although it is indicated in Gt units.

Thanks for pointing out this interesting issue. All the quantities in Eqs (1) and (2) are fluxs in Gt/yr. We have changed delta-m to delta-m/delta-t, to make it a flux.

In short, we have updated the manuscript to make more clear that Eqs.(1-2) refer to fluxes (in Gt per time unit). We add that
"The quantities in Eq. 1 refer to fluxes (in Gt per time unit)."; "The units are in Gt per time unit." (For Eq. 2).

As for Fig. 9 (now Fig. 5 in the revised manuscript), it shows *monthly* meltwater production, so that we believe that showing the results in Gt is fine.

TECHNICAL DETAILS
- In the abstract mass losses are reported in terms of negative numbers. A negative mass loss is a mass increase.

Thanks for pointing this out. We have changed "mass loss" to "mass variations".

- A hyphen is not a minus sign. Please write a minus sign, wherever it should be a minus sign, including in the legends and captions of graphs and in headers, etc.

Done.

- The acronym DS is not very helpful. Write out in full everywhere.

Changed as suggested.

- Red pentagrams are not very visible in figure 1. Figure 1 can certainly be improved. Explain the blue patches outside Greenland briefly in the caption.

Done. The blue patches outside Greenland are briefly explained. The red pentagrams are changed to "×". We admit, since the locations of glaciers are quite close, making it difficult to identify. (Note that those glaciers are exactly the set of glaciers discussed in Moon et al. (2014).)

- Page 6, line 25: Least-squareS adjustment. (If it were singular, there is nothing to adjust).

Done.

- Be consistent in your mathematical typesetting. Take, e.g., the symbol "f" for flux gate in eqn (3) and in the line above. In the equation there are two different letters "f" and in the line above, the "f" should be set in math italic. Similarly, in line 24 (page 6), the v should be bold-math-italic. And check the N and i in the last line of page 6.

Done.

- Several graphs show mass variation with the unit (EWH: m). That is inappropriate. The quantity is (expressed in) EWH and its unit is m.

We have changed the y-axis to "EWH (m)".

- Page 26, Line 5: The sentence "No regularization is applied..." is preceded by an explanation of truncated eigenvalue decomposition. Now that is definitely regularization.

We agree with the reviewer that the truncation of the eigenvalues is also a kind of regularization. However, it was the error covariance matrix subject to the truncation, and not the normal matrix. What we mean with the statement "No regularization is applied..." is that there is no spatial regularization applied to the normal matrix in the lease-squares adjustment. We have made it clearer in the main text.

---

## Author Response (AR2)

**Responses to the editor**

We are grateful for the comments provided by the editor. Please find below our answers in red to the editor's comments in black and the suggested changes in the MS main text in red.

On behalf of all authors,

Jiangjun Ran

Comments to the Author:
Dear Drs. Ran et al.,

Thanks for submitting your manuscript to The Cryosphere Discussions and your thorough response to the two reviewers' comments. I am generally satisfied with both your responses and revisions to the manuscript. The arguments are clearer and now better focused on the most significant and defensible elements of your seasonal mass-anomaly analysis, rather than the mass trend or the areas with seasonal signals that aren't yet clearly significant. The latter half of the manuscript is particularly strong now. Below I have several minor comments for consideration prior to a final decision on your manuscript.

We appreciate the insightful comments by the editor. We have changed the manuscript accordingly. For more details, please see the text below.

The abstract still misses the mark a bit and doesn't highlight the key discovery from this study as well as the title does. I have several suggestions for it:
- If appropriate (I believe it is), add "and previous studies" to the end of the sentence that starts with "This estimate…".

Done.

- For the sentence that presently starts with "Most importantly,", I suggest a revision along the lines of: "We further identify a seasonal mass anomaly throughout the GRACE record that peaks in July at 80–120 Gt and which we interpret to be due to a combination of englacial and subglacial water storage generated by summer surface melting.

Done. Thank you very much for this suggestion, which makes the abstract better hightlight the key discovery from this study.

- Add a concluding sentence on what future studies could do be better understand the seasonal mass anomaly or its broader significance in the glaciological investigation of the GrIS. This is important to connect with the second sentence of the abstract, which refers to needing a better understanding of "mechanisms driving the observed mass loss" but the rest of the abstract doesn't really address that need.

Done. We add

"With the improved quantification of meltwater storage at the seasonal scale, we highlight its importance to understand glacio-hydrological processes and their contributions to the ice sheet mass variability."

- 1/10: SE and NW abbreviations not needed or then used in abstract.

Done. We have removed the abbreviations.

- 1/12: "Gt" should be spelled out if not associated with a specific value.

Done. Thanks.

Other minor comments:
1/15: "(0.4–1.2)" range seems unnecessary here.

Done.

1/17: Already defined GrIS on 1/14.

Done.

2/12-14: Reword the new sentence as it is awkward and unclear as presently constructed.

Done. We changed to
"Importantly, ice flow velocities have increased during the last decade and shown different spatial and temporal patterns (Moon et al., 2012)."

2/21: "…a few marine-terminating glaciers were investigated by, e.g., Howat et al…"

Done.

2/24: I suggest changing this first sentence to: "GrIS mass balance also depends on supra-, en- and subglacial meltwater storage".

Done.

2/26: Change to: "However, time-varying total englacial and subglacial meltwater retention…has been poorly quantified…".

Done.

2/32: "So far, no attempt to quantify…"

Done.

3/13: "components" not "compartments".

Done.

6/13: "Previous studies on the…"

Done.

Figure 2: X-axis label is wrong. Should simply be "Year CE" rather "Time (Yr)". Same applies to all other figures whose x axes cover multiple years, e.g., Figure 10.

Done.

9/21: Mass anomalies or mass change anomalies? Unclear

Done. We have changed it to "mass change anomalies".

9/22: "…contributes 75% of the total acceleration…"

Done.

9/25-26: "However, we note that the…"

Done.

10/1: "…we refer the reader to…"

Done.

Figure 7 caption: Not necessary to mention unit at end of caption.

Done.

Figure 8: Stack these three panels to make better use of space.

Done. We did the same for Figure 7. Thanks.

20/7: "…revealed the presence of the short-term…"

Done.

Throughout the manuscript:
- Gt yr–1 not Gt/yr.

Done.

- Spell out "Gt" as "gigatonnes" when not assigning the unit a specific value, e.g., 13/29.

Done.

- No need to hyphenate "englacial" or "subglacial" except "en-" on 2/2.

Done.

- 2/5 and elsewhere: "(mass per unit time)" not "(Gt per time unit)" or "(mass per time unit)".

Done.

- 2/15: Am I correct in thinking that "intra-annual" and "seasonal" effectively mean the same

thing here? If possible, it would be good to select one and use throughout. IMO, "seasonal" is more evocative and correct.

Done. We have unified to use "seasonal". Thanks.

- 2/24: "retention" and "storage" are also effectively synonyms here, so it would be best to pick one and stick with it.

Done. We choose to use "storage". Thanks.

- For all figures that showing months with all or a portion of the seasonal cycle, it would be more accurate visually to show the seasonal patterns at the middle of the month rather than its beginning, given that they are monthly averages. Further, the x axis label should then be month abbreviation instead of numbers, since month names are referred to in the text, not month numbers. For example, for Figure 6: A M J J A S O N instead of 4-11, with the labels and values shown at 4.5-11.5 instead of 4-11 and across a new range of 4-12.

Done.